# Interdisciplinary Care Networks in Rehabilitation Care for Patients with Chronic Musculoskeletal Pain: A Systematic Review

**DOI:** 10.3390/jcm10092041

**Published:** 2021-05-10

**Authors:** Cynthia Lamper, Laura Beckers, Mariëlle Kroese, Jeanine Verbunt, Ivan Huijnen

**Affiliations:** 1Department of Rehabilitation Medicine, Functioning, Participation & Rehabilitation, Care and Public Health Research Institute (CAPHRI), Faculty of Health, Medicine and Life Sciences, Maastricht University, 6229 ER Maastricht, The Netherlands; laura.beckers@maastrichtuniversity.nl (L.B.); jeanine.verbunt@maastrichtuniversity.nl (J.V.); ivan.huijnen@maastrichtuniversity.nl (I.H.); 2Department of Health Services Research, Care and Public Health Research Institute (CAPHRI), Faculty of Health, Medicine and Life Sciences, Maastricht University, 6229 ER Maastricht, The Netherlands; marielle.kroese@maastrichtuniversity.nl; 3Centre of Expertise in Rehabilitation and Audiology, Adelante, 6432 CC Hoensbroek, The Netherlands

**Keywords:** interdisciplinary care, rehabilitation care, care networks, Quadruple Aim, health, experienced quality of care, satisfaction with work, cost-effectiveness, primary care

## Abstract

This systematic review aims to identify what rehabilitation care networks, within primary care or between primary and other health care settings, have been described for patients with chronic musculoskeletal pain, and what their impact is on the Quadruple Aim outcomes (health; health care costs; quality of care experienced by patients; work satisfaction for health care professionals). Studies published between 1 January 1994 and 11 April 2019 were identified in PubMed, CINAHL, Web of Science, and PsycInfo. Forty-nine articles represented 34 interventions: 21 within primary care; 6 between primary and secondary/tertiary care; 1 in primary care and between primary and secondary/tertiary care; 2 between primary and social care; 2 between primary, secondary/tertiary, and social care; and 2 between primary and community care. Results on impact were presented in 19 randomized trials, 12 non-randomized studies, and seven qualitative studies. In conclusion, there is a wide variety of content, collaboration, and evaluation methods of interventions. It seems that patient-centered interdisciplinary interventions are more effective than usual care. Further initiatives should be performed for interdisciplinary interventions within and across health care settings and evaluated with mixed methods on all Quadruple Aim outcomes.

## 1. Introduction

Chronic musculoskeletal pain (CMP) is a leading cause of disability occurring in 19–28% of the European population [1,2]. As many as one-third of primary care consultations concern CMP complaints such as back or knee pain [3]. Complaints often persist for more than five years and have significant impacts on patients’ daily life, leading to high societal and health care costs [4,5,6].

Research indicates that pain needs an integrated biopsychosocial approach to decrease its impact on health. Nevertheless, this impact is expected to increase as people live longer [7,8]. Current care is organized in “silos” with a focus on only one aspect of pain (biomedical, psychological, or societal), instead of an integrated approach. There is little coordination and communication among health care professionals (HCPs), leading to fragmented care [9,10,11]. This results in many monodisciplinary treatments, with a wide variety of treatment approaches, restricted in available time and resources. Thus, there is a call for a different organization of care for patients with CMP [12]. As a possible solution for this fragmented care, the general practitioner (GP) should have a more prominent role in managing patients with chronic and complex diseases such as CMP [13], as there is a need for continuity, comprehensiveness, and coordination in CMP care [14]. Primary care should play a central role in effectively managing and integrating care with mono- and multidisciplinary treatments, with the GP as the case-manager for chronic and complex diseases [15].

Accordingly, the World Health Organization (WHO) developed a guideline for redesigning rehabilitation in health systems [16]. It indicates that rehabilitation services should be integrated within primary care, as well as between primary, secondary, and tertiary levels of health systems, with a case-management role for primary care. This is in line with Coleman et al., who state that disease management interventions that target only patients may be less effective than those that also focus on organization of care and redesign of care delivery [9]. Cieza et al. advise scaling up rehabilitation services in primary care worldwide to ensure that a life-course and integrated perspective on care is achieved [2]. CMP must be approached within a biopsychosocial framework in order to deliver the most effective treatment. Depending on the complexity of the pain problem, it must be provided by different health care disciplines in collaborative teams, either in primary care alone or combined with secondary and tertiary care [17,18].

Collaborative teams of HCPs for pain management in primary care could range in scope from less extensive combinations of GPs and HCPs, focusing on physical and psychological aspects of pain, to broad teams including rehabilitation, psychology, nursing, and case management [19,20]. If these treatments are multimodal, meaning one therapeutic aim per discipline, and involve HCPs from different disciplines, they are termed “multidisciplinary” [21]. For patients with more complex complaints, an interdisciplinary treatment is needed, where all HCPs involved have a common therapeutic aim and a shared biopsychosocial focus. These interdisciplinary care networks can improve clinical care and service delivery, as suggested by Coleman et al. [9].

In order to optimize health system performance, including interdisciplinary care networks, an approach known as the Quadruple Aim is recommended. This comprises four dimensions: health, quality of care experienced by patients, healthcare costs, and HCP work satisfaction [22,23]. There is some evidence that interdisciplinary care networks designed for various diseases can improve these four Quadruple Aim pillars [24,25,26]. Moreover, interdisciplinary care networks for CMP, incorporating a biopsychosocial model in assessing and treating pain, can result in pain reduction, improved quality of life, and improved social functioning [20]. In some cases, return-to-work and vocational outcomes may be seen.

However, it is not known which biopsychosocial interdisciplinary care networks exist within primary care, or between primary care and other health care settings, for the rehabilitation of patients with CMP. Furthermore, the impact of such networks on the Quadruple Aim outcomes of health, health care costs, quality of care experienced by patients, and work satisfaction of HCPs, is unknown. Therefore, this study aims to address these uncertainties. The first research question is: Which interdisciplinary care networks within primary care and between primary care and other health care settings have been implemented in rehabilitation care for patients ≥18 years with CMP over the last 25 years? The second question is: What is the impact of these interdisciplinary care networks on rehabilitation care for patients with CMP, in terms of the Quadruple Aim outcomes: health, health care costs, quality of care experienced by patients, and HCP work satisfaction?

## 2. Materials and Methods

We conducted a systematic literature review to synthesize studies with interdisciplinary care networks in care for patients with CMP. These interdisciplinary care networks must be implemented within primary care or between primary care and other healthcare settings (secondary or tertiary care, social care, or community-based care) (see Figure 1). As this research did not involve human subjects, we did not seek ethics clearance for the project. The protocol was registered in the international prospective register of systematic reviews (PROSPERO; http://www.crd.york.ac.uk/PROSPERO/ (accessed date: 6 May 2021)) on 28 August 2020 (registration number CRD42020158057). The review was conducted following PRISMA guidelines [27,28].

### 2.1. Databases Searched and Inclusion and Exclusion Criteria

Studies published between the 1 January 1994 and the 14 November 2019 were identified by searching the databases PubMed, CINAHL, Web of Science, and PsycInfo, and tracing publications from the reference sections of included papers and relevant reviews. Studies were included if the main population comprised patients with CMP, the intervention was implemented in primary care, or a combination of primary care and other health care settings, with a rehabilitation aim and an interdisciplinary care network. Only original descriptions of interventions in Dutch, English, or German were included. Detailed definitions of the in- and exclusion criteria can be found in Table 1. In addition to these criteria, studies with populations comprising a mix of patients with subacute and chronic complaints or with a non-disease-specific intervention were included. From these studies, only the results for patients with CMP were taken into account. Studies investigating group interventions delivered at the same time by HCPs of different disciplines were also included in this review as it was assumed that they would have discussed treatment approaches. In all interventions, the collaboration between HCPs had to be bidirectional to be included. When other articles described the same intervention, these were also included in our review.

### 2.2. Search Strategy

Terms used were defined by scoping searches and team discussions. An information specialist finalized the search strategy and adapted keywords according to the configuration of each database. Our search strategy has been published online in detail (Appendix A). Briefly, it included variations on the following terms: ‘*chronic musculoskeletal pain*, *fibromyalgia*, *regional pain*, *arthritis*, *interdisciplinary*, *integrated*, *multidisciplinary*, *service*, *system*, *delivery*, *network*, *physical and rehabilitation medicine*, *Quadruple Aim*, *health outcome*, *quality of care*, *healthcare costs and satisfaction with work*’. Three reviewers (CL (100%), WM (75%), and LB (25%)) independently screened title and abstract, and two reviewers (CL (100%) and LB (100%)) screened all full texts. Disagreements were solved by an arbitrator (IH). Identified references were downloaded and collected using EndNote bibliographic software (Clarivate Analytics, Philadelphia, PA, USA), and the article selection was performed in the review processing software, Rayyan [29].

### 2.3. Data Extraction and Analysis

CL extracted data on the interdisciplinary care networks, study aims, and outcomes of the included articles. LB reviewed 25% of the data extraction. First, descriptions of the included interventions were compiled. These included: country; name of intervention; target population; health care setting; and description of the collaboration and intervention. If multiple articles were published for one intervention, these were merged to give a complete overview. As shown in Figure 1, health care settings were classified based on the type of intervention: within primary care or between primary care and secondary or tertiary care, community-based care, and/or social care. Interventions in secondary or tertiary care were combined as one category because the distinction between secondary or tertiary care was not always clear from the descriptions given. Descriptions of interventions were extracted from the studies and classified into these categories:Assessment (a systematic approach to ensuring that the health service uses its resources to improve the health of the population most efficiently) [31];Education—basic knowledge (anatomy, biomechanics, the function of the body, and pathophysiology) [32];Education—knowledge of disease prevention and ergonomics (information on prevention, cause of pain, ergonomics, information on posture, information on activity, exercise) [32];Education—knowledge of treatment (self-management, lifestyle modification, information on coping with the problems) [32];Manual Therapy (passive joint mobilization and massage therapy) [33];Specific Exercise Therapy (active and/or active-assisted strengthening, mobilizing, and stretching exercises to restore the function of the affected region) [34];General Exercise Therapy (aerobic and resistance training, causing an increase in energy expenditure, to maintain health-related outcomes) [35];Mind-Body Exercise Therapy (to enhance the mind’s capacity to positively affect bodily functions and symptoms, including pain, by combining exercises with mental focus) [36];Cognitive behavioral therapy (CBT) (integration of exercise therapy with daily performed activities based on cognitive-behavioral principles, time-contingent) [37,38,39,40,41];Workplace intervention (a set of comprehensive health promotion and occupational health strategies implemented in the workplace to improve work-related outcomes) [42];Anesthetics (local anesthetics for diagnosis and therapy, indications include functional disorders, inflammatory diseases, and acute and chronic pain) [43];Medication management (a systematic process of ensuring that the patient’s medication regimen is optimally appropriate, effective, and safe, and that the patient is adhering to this regimen to promote health and reduce the need for health care use) [44].

Second, outcomes relevant to the Quadruple Aim were extracted for each intervention. For these, study dates, study designs, outcome measures with measurement instruments for relevant primary outcomes, and results were recorded. The results of randomized trial designs were presented in two categories: either (1) positive and significant (+) (*p* < 0.05) compared to the comparator intervention for randomized controlled trial (RCT) designs; or (2) positive and non-significant (*p* > 0.05), no difference between the intervention and comparator intervention, or alternatively negative and significant for the intervention compared to the comparator intervention (-) (*p* < 0.05). For non-randomized trial designs, results were classed as significant (+) (*p* < 0.05) or non-significant (-) (*p* > 0.05), compared to baseline. Mixed positive and negative results for subdomains are indicated by +/-. For the qualitative studies, opinions are summarized as all positive (+), negative (-), neutral (=), or mixed (+/-). All outcomes from the studies relevant to the Quadruple Aim are presented. Primary outcomes of the studies were identified using the following procedure. First, the primary outcome, as described by the authors, was chosen. If this was not described, the outcome measure used in the sample size calculation was chosen. If this was also not described in the article, the outcome measure best fitting the aim of the intervention was chosen (e.g., aim: improving functioning, outcome measure: health-related quality of life or functioning; aim: return to work, outcome measure: return to work or sick leave; etc.). In the case where this was also unclear from the article, the first choice was the outcome measure for quality of life (often measured in this type of study, making it comparable). Then, costs such as sick leave or return to work were the second choice. After that, quality of care experienced by patients or HCP work satisfaction were the third choice. Based on the homogeneity and the chosen outcome measures, a meta-analysis was considered.

### 2.4. Risk of Bias in Individual Studies

Quality assessment tools specific to the method(s) employed were used. These tools were used to assess and compare the quality of RCT designs, non-randomized study designs and qualitative designs. For RCT designs, the risk of bias was assessed using Version 2 of the Cochrane risk-of-bias tool for randomized trials (RoB2 tool) [45]. The Risk of Bias in Non-randomized Studies-of Interventions (ROBINS-I) tool was used for assessing non-randomized study designs [46]. Domains not relevant for studies without a control group are marked (-) or (NA). For qualitative designs, the Joanna Briggs Institute Critical Appraisal Tools, Checklist for Qualitative Research, was used for a critical appraisal [47]. This critical appraisal tools assist in assessing the trustworthiness, relevance and results of published papers. At least seven questions (out of 10) had to be answered “yes” to receive a positive overall appraisal. Articles describing study protocols were not assessed for risk of bias. One researcher (CL) assessed the risk of bias and performed the critical appraisal for each study. One researcher (LB) randomly cross-checked 25% of the included studies. Disagreements were resolved by an arbitrator (IH). Review authors were not blinded for author names, institutions, or journals. If additional information was needed, corresponding authors would have been contacted. Results are reported through graphical representation of bias judgements grouped by design of study.

## 3. Results

The process of the literature review is shown in Figure 2. The combination of keywords yielded 15,428 potentially relevant articles in the databases on 14 November 2019. Overall, 2926 articles were excluded by deduplication, and 11 studies were added with the snowball method, resulting in 12,513 articles. After reviewing the titles and abstracts of the articles, 12,152 of these were excluded. In total, 361 full-text articles were assessed for eligibility, and 320 were excluded for not meeting one or more of the criteria. The most common reasons for exclusion were interventions without an interdisciplinary care component or interdisciplinary interventions without a role for primary care. If interventions were described in other articles as protocols or allied studies, these were also included, leading to 49 included articles describing 34 interventions.

### 3.1. Overview of Included Studies

An overview of the included interventions is presented in Table 2. Of the 34 included, 21 consisted of a collaboration of HCPs within primary care [25,48,49,50,51,52,53,54,55,56,57,58,59,60,61,62,63,64,65,66,67,68,69,70,71,72,73,74,75,76]. Examples of these were collaborations between therapists (TH) and nurse practitioners (NP) or more extensive collaborations between physicians/physiatrists (PH), psychologists (PSY), and various THs. These collaborations ranged from merely performing an assessment to giving a complete interdisciplinary treatment in primary care. Nearly all interventions included at least one education module. Only three had a medication management module. Moreover, most studies with collaborating THs had general or specific exercise therapy modules in the intervention. If the interventions had a PSY or Psychosocial counsellor (PSY-C) in the team, these interventions were often focused on Mind-Body Exercise Therapy or CBT.

Furthermore, six interventions existed of a collaboration between primary care and secondary or tertiary care [48,49,50,51,52,53]. Two of these interventions were between a rehabilitation department and primary care [48,51]. Examples of these collaborations were a GP in primary care with a TH, orthopedic surgeon/specialist (OS), or NP in secondary or tertiary care. Collaborations between GPs and extensive rehabilitation teams, consisting of a nurse (NRS), PSY, TH, or PSY-C, and OS. Four interventions existed of an interdisciplinary assessment. In one study, assessment and follow-up were performed by an HCP in secondary care, a TH and a patient via video-conferencing (due to long-distance) [53]. Rothman et al. evaluated a collaboration in assessment and giving advice between a GP and at least three HCPs in secondary or tertiary care [52]. Two of these interventions consisted of an interdisciplinary assessment followed by treatment [48,51]. The other two interventions consisted of an interdisciplinary treatment without an interdisciplinary assessment [49,50].

Additionally, one intervention was applied in an interdisciplinary pain clinic in primary care with collaboration in primary care, as well as between primary care and secondary and tertiary care [54]. THs who usually work in both primary care and secondary/tertiary care settings delivered the treatment in this interdisciplinary pain clinic in primary care, consisting of specific exercise therapy, medication management, and education.

Two interventions were a collaboration between primary care and social care [55,56]. Here, the teams consisted of several THs, a PH, and a case manager. Together, they performed a team assessment and, during the treatment and follow-up meetings, combinations of the HCPs involved delivered the treatment. In both treatments, workplace interventions were included, aimed at a return to work.

In addition, two interventions consisted of a collaboration between primary care, secondary/tertiary care, and social care [57,61]. These extensive interventions also involved the patients’ medical specialists during workplace interventions, in addition to THs, GPs, and occupational physicians (OPs). While both interventions had many similarities, the recruitment of the study populations differed. In the studies of Steenstra et al. and Anema et al., the participants were recruited by the OP, while in the studies of Lambeek et al., the recruitment was by the PHs of the outpatient clinics of participating hospitals [57,59,60,61].

Finally, two interventions existed of a collaboration between primary care and community-based initiatives [62,66]. In the intervention of McBeth et al., & Bee et al., the TH delivered the CBT, and the fitness instructor (FI) from a community-based initiative gave the general exercise therapy [62,66]. The intervention of Bennell et al., and Hinman et al. comprised a physical therapy program delivered by the TH in primary care, CBT by the telephone coach (TC), and an information booklet for education about the disease [64,65].

### 3.2. Quadruple Aim Outcomes

An overview of the Quadruple Aim outcomes for each intervention is presented in Table 3. After data extraction from the included studies, it became evident that the interventions, outcome measures, and study designs were too heterogeneous to justify meta-analysis in the included studies. Therefore, narrative analyses were conducted.

Among the 49 articles, 19 randomized trials, 12 non-randomized studies, 7 qualitative studies, 7 study protocols, 1 description of an intervention, 2 studies with a population with mixed diagnoses, and 1 study regarding barriers and facilitators, were found. Thirty-nine articles had at least one of the Quadruple Aim outcomes as the primary outcome: 18 articles described health outcome measures, 12 described cost outcome measures, 4 described quality of care experienced by patients, and 5 articles describe work satisfaction for HCPs. Hinman et al. described quality of care experienced by patients and HCP work satisfaction as the combined primary outcome [65]. Most studies measured more than one Quadruple Aim outcome, but only two interventions intended to assess all Quadruple Aim outcomes. Dobscha et al. measured all Quadruple Aim outcomes but presented only the baseline results [67]. Bath et al. described all Quadruple Aim outcomes in the protocol article, but not all results are published yet [68].

The outcomes of Dobscha et al., and Gustavsson et al., were only described as baseline measurements for an RCT [67,69]. From the remaining articles, comprising study protocols, description of the intervention, studies with mixed diagnoses, and the study regarding barriers and facilitators, no outcomes could be extracted [25,48,51,52,53,54,58,60,71,72,83,84,85,96]. These studies were used for descriptions of interventions in Table 2.

#### 3.2.1. Within Primary Care—Randomized Trial Designs

The most frequently presented outcome measure among randomized trial designs was pain intensity (five studies [70,71,72,73,74]). For Helminen et al., the maximum follow-up time was three months, resulting in a non-significant difference between intervention and control groups [73]. Four studies reported outcomes at one-year follow-up: two reported a significant improvement (50%) [71,72], while two reported no significant improvement (50%) [70,74]. Pain intensity scores were measured with the 100-mm Visual Analogue Scale (VAS), the Chronic Pain Grade Severity subscale, or the 0–10 numerical pain rating scale (NPRS). Health-related quality of life (HRQoL) was measured in five studies. The improvement on the VAS score at six months of Hansson et al. (2010) was significantly different between intervention and control groups (20%) [75]. In the other four studies (80%), HRQoL was not different between the groups at three, four, or twelve months, as measured with the RAND-36, Short Form-36, and EQ-5D questionnaires [70,72,73,76].

Outcomes regarding sick leave/working days or medication prescription and use were most often measured for the Quadruple Aim of health care costs (five studies). At four-month follow-up, Calner et al. found positive and negative changes at the different levels of working percentages for the number of working participants in the intervention group compared to the control group [70]. Regarding sickness absence, Gustavsson et al. found no significant change in absence at one-year follow-up [69]. Additionally, Helminen et al. found no significant difference in the number of sick-leave days at three-month follow-up between the intervention and control groups [73,76]. They also found no difference in medication use at three-month follow-up between the control and intervention group, just like Sundberg et al. at four-month follow-up [73,76]. Dobscha et al. found a significant difference in opioid prescriptions at one-year follow-up but non-significant differences in the use of adjuvant pain medications between intervention and control groups [72].

Quality of care experienced by patients was measured in two studies. On a self-assessment questionnaire measuring the quality of care completed by patients, Chelimsky et al. found a significant result at one year for the facilitation of patient involvement in care, though no differences at nine weeks and one year were seen by Gustavsson et al. [69,71].

Both of these studies also examined HCP satisfaction with the care they delivered. At one-year follow-up, HCPs providing the interventions rated their work (significantly) more positively than did HCPs in the control conditions.

#### 3.2.2. Within Primary Care—Non-Randomized Study Designs

All included studies had a longitudinal design (follow-up after intervention), but only two studies included a control condition. All seven interventions evaluated health outcomes. Pain intensity was only reported in two studies, with mixed results at one year and significant pain decreases at five years in both studies [87,89]. In four studies, including one protocol, HRQoL was the primary outcome, and in one study, it was the secondary outcome. At three months, six months, one year, and five years, significant changes were found. HRQoL was measured with the Short Form-36 or a self-constructed questionnaire [86,87,90]. Three studies reported no change in quality of life at one- and three-year follow-ups, as measured with the Short Form-36, EQ-5D, or self-constructed questionnaire [87,89,90].

Costs were evaluated in six interventions, with sick leave or paid work participation as the most frequently reported measurement. In Dunstan et al. and Westman et al., the changes in paid work participation and sick leave were non-significant (40% of the studies) at six-months and three-year follow-up, respectively [80,90]. Stein et al. and Mårtensson et al. found at one- and two-year follow-ups significant changes, compared with baseline assessment (40% of the studies) [84,87]. Westman et al. reported mixed results (20% of the studies) for both sick leave and return to work [89]. Stein et al. and Westman et al. found no significant changes at one- and three-year follow-ups for opioids and drug consumption [87,90]. On the other hand, Gurden et al. found a significant decrease in medication usage after discharge [82].

Three studies found significant positive (100%) results for quality of care experienced by patients [82,86,89].

None of these interventions evaluated HCPs’ satisfaction with the care they delivered.

#### 3.2.3. Within Primary Care—Qualitative Designs

None of the six qualitative designs within the primary care interventions evaluated changes in health or costs.

Quality of care experienced by patients was assessed on different items by three studies, with mixed results [79,81,93]. For example, Dunstan et al. found that patients made both positive and negative points about the usefulness of the program for managing their pain, helping them to become more active and to get back to work, but expressed only positive views about the quality of treatment by HCPs. [81]. Additionally, one study found clear positive results (atmosphere, environment, value of one’s contribution) and negative results (expectations of a sick person, reacting but not acting, awareness and integration) on various items [85]. Lovo et al. reported positive results for the quality of care, measured with both qualitative questionnaires and interviews [92].

Lovo et al. evaluated HCP work satisfaction, reporting overall positive results regarding access to care, effective inter-professional practice, and enhanced clinical care [92]. Only technology (telehealth) was scored less positively.

#### 3.2.4. Between Primary Care and Secondary or Tertiary Care—Randomized Trial Designs

All three randomized trial designs of the interventions combining primary care and secondary or tertiary care measured health outcomes [50,52,53]. Rothman et al. reported a non-significant difference between the intervention and control groups at 15-month follow-up for the primary outcome of pain intensity [52]. In the study of Taylor-Gjevre et al., the difference between intervention and control groups in disease activity at a nine-month follow-up was also non-significant [53]. All three studies measured HRQoL but different measurement instruments were used (EQ-5D, Short Form-36, and a self-constructed questionnaire). In two studies, significant results were found on some questionnaires’ subscales, with non-significant results on other subscales, at one-year and 15-month follow-ups, while one study found non-significant results at a nine-month follow-up.

Cost outcomes were measured by Haldersen et al., and Rothman et al. [50,52]. Changes in return to work after 12 months did not differ between groups, while changes in ability to work did at a 15-month follow-up.

Rothman et al., and Taylor-Gjevre et al. measured experienced quality of care by patients with questionnaires [52,53]. Although no differences were found at nine months, at the 15-month follow-up, the intervention group rated quality of care experienced higher than did the control group. HCP work satisfaction was not measured in any of these three studies.

#### 3.2.5. Between Primary Care and Secondary or Tertiary Care—Non-Randomized Trial Designs

Health outcomes were assessed with less widely used outcome measures by all three non-randomized trial designs [48,49,51]. Burnham et al. had pain interference as the primary outcome, which was significant after treatment discharge in the cohort study [48]. Plagge et al. found significant changes in all domains of HRQoL post-intervention [51]. Claassen et al. found significant improvement only in illness perceptions after three months, whereas non-significant differences were found for BMI, pain, and limitations in functional activities, and physical activity after three months [49].

The study of Claassen et al. was the only one measuring the health care costs and quality of care experienced by patients, goals of the Quadruple Aim; both had significant results at the three-month follow-up [49].

None of the three included studies measured HCP work satisfaction.

#### 3.2.6. In Primary Care and between Primary Care and Secondary or Tertiary Care—Randomized Trial Design

Only one study was included which described a collaboration within primary care as well as between primary care and secondary or tertiary care [54]. The study had grip strength as the primary outcome, which was significantly different in the intervention group at the two-month follow-up. The other health outcomes, pain and health status, showed non-significant differences at that time point.

This study also found significant results regarding quality of care experienced by patients at the two-month follow-up.

Costs and HCP work satisfaction were not measured.

#### 3.2.7. Between Primary Care and Social Care—Randomized Trial Design

Bültmann et al. was the only included study evaluating collaboration between primary and social care [55]. For health outcomes, no significant changes in pain intensity and functional disability between intervention and control groups were reported at any time point.

Cumulative sickness absence hours was the primary outcome of this study. Results showed a significant decrease in sick leave at six and 12 months, compared to the control group.

Bültmann et al. did not measure outcomes on quality of care experienced by patients or HCP work satisfaction [55].

#### 3.2.8. Between Primary Care and Social Care—Non-Randomized Study Design

Heijbel et al. evaluated the collaboration between primary and social care, but, in this study, health outcomes and quality of care experienced by patients were not measured [56].

For return to work, the primary outcome of this study, a significant improvement was found after two years.

Mixed results were reported regarding HCP work satisfaction, measuring the experiences of executing and implementing a workplace-based rehabilitation intervention.

#### 3.2.9. Between Primary Care and Secondary or Tertiary Care and Social Care—Randomized Trial Designs

Two interventions in this group were evaluated, one intervention by two studies and the other by one [57,59,60]. Two studies measured health outcomes: pain intensity with the VAS and functional status with the Roland-Morris Disability Questionnaire [57,60]. Only functional status was positively changed at 12 months, whereas the other associations at 3, 6, and 12 months were all non-significant between the groups.

In all three studies, the primary outcome was the duration of sick leave to a full return to work. In both studies of Lambeek, significant differences were found at the 12-month follow-up, favoring the intervention over the control group [59,60]. However, in the study of Anema et al., non-significant differences between interventions were found within the same timeframe [57].

No measurements were performed for the quality of care experienced by patients or HCP work satisfaction.

#### 3.2.10. Between Primary Care and Community-Based Care—Randomized Trial Designs

Two interventions, with three randomized trial designs, of which one was a protocol, were found reporting such collaboration between primary care and community-based care. Two studies measured health outcomes [63,66]. McBeth et al. had change in health as the primary outcome, which was significantly more improved at six and nine months than in the control group [66]. Significant differences in changes were also found between both conditions for kinesiophobia at nine months and for the physical component of HRQoL at both time points, measured with the Short Form-36. For the mental component score, general health and chronic pain grade, no significant results were found. Bennell et al. found non-significant differences in measured health outcomes, of which knee pain intensity and physical functioning in the previous 48 h were the primary outcomes [63].

Costs were assessed in the study of McBeth et al. with a cost-effectiveness analysis [66]. Non-significant differences between intervention and control group at the six- and nine-month follow-ups were found.

No measurements were performed for the quality of care experienced by patients or for HCP work satisfaction.

#### 3.2.11. Between Primary Care and Community-Based Care—Qualitative Designs

Bee et al., and Hinman et al. performed qualitative evaluations for Quadruple Aim goals [62,65]. Mixed results were found regarding participants’ illness experiences, which was a health outcome [62]. No cost outcomes were assessed qualitatively.

Quality of care experienced by patients had mixed results in the study of Bee et al. [62]. Positive as well as negative results were found regarding treatment preferences and the perceived fit with the interventions and their patients’ needs. Hinman et al. measured the satisfaction of HCPs in combination with experiences of patients [65]: their only positive results regarded HCPs’ interest in patients during the treatment and in collaboration. Mixed results were found regarding information and accountability, program structure, and roles and communication in teamwork.

### 3.3. Risk of Bias

The results of the risk of bias (RoB) per domain for the randomized trial designs (*n* = 19) are presented in Figure 3 and for the non-randomized study designs (*n* = 12) in Figure 4. The results of the critical appraisal of the qualitative designs (*n* = 7) can be found in Figure 5.

### 3.4. Randomized Trial Designs

Overall, the studies of Dobscha et al., and Lambeek et al., were found to have a low RoB [60,72]. Ten studies raised some concerns in the RoB, and seven studies had a high RoB. The randomization process had a low RoB for 15 studies, while four studies had some concerns [53,57,69,71]. Deviations from the intended interventions most often led to there being some concerns for RoB, in most cases because participants, patients and HCPs, were unblinded. Chemelinsky et al. and Halderson et al. were judged to have some concerns for this RoB domain [50,71]. Completeness of outcome data and measurements of outcomes most often had a low RoB, though there some concerns for three studies [54,73,95]. Six studies had a high RoB on these domains [55,60,66,69,70,71]. There were some concerns about possible selection of reported results with fourteen studies because most study protocols were not published, so planned outcomes could not be matched with published outcomes [49,50,52,53,54,55,66,67,69,70,71,74,75,76].

### 3.5. Non-Randomized Study Designs

The studies of Dunstan et al., and Heijbel et al. did not report enough information to assess the RoB [56,80]. All other studies were assessed as at high RoB [48,49,50,51,56,80,82,83,84,86,90]. The main reason was the lack of a control group in these studies, which indicated a serious RoB in the domain of confounding. The domain classification of the interventions was not applicable for studies without a control group. Moreover, this domain was found to have a serious RoB because outcome measures could be influenced by the outcome assessors and/or unblinded patients in most studies. The lack of publication of a protocol in any of these studies led to an absence of information about any bias in the selection of the reported results.

### 3.6. Qualitative Designs

Six studies scored positively in the critical appraisal of study methods [62,65,79,85,92,93]. The study of Dunstan et al. was described only briefly, hindering assessment, and so this study scored negatively [81]. In most studies, the philosophical perspective was not described and a statement locating the researcher culturally or theoretically was not made.

## 4. Discussion

As far as we know, this systematic review is the first to identify which interdisciplinary rehabilitation interventions have been described within primary care, and between primary care and other health care settings, delivering rehabilitation care to patients with CMP. In addition, we describe the impact of these interdisciplinary interventions in rehabilitation care for patients with CMP, in terms of the Quadruple Aim goals: health, quality of care experienced by patients, health care costs, and HCP work satisfaction. The review was based on 49 articles (34 separate interventions), including 19 randomized trials, 12 non-randomized studies, 7 qualitative studies, and 11 articles with a description of the intervention but without a description of relevant outcomes and/or results.

In summary, of the studies that examined *interventions situated in primary care (n = 19)*, most did not find significantly improved health outcomes compared to care as usual. In the non-randomized designs, in general, health outcomes seemed to improve over time. However, cost outcomes and quality of care experienced by patients in intervention groups showed a mixture of significant improvements and non-improvements in both randomized and non-randomized trial designs. The differences in satisfaction levels may relate to the fact that new interventions can change the usual care pathways, which could be challenging for the patient to follow and thus lead to lower satisfaction. Alternatively, patients who were already very satisfied with the current regular health care received might experience little or even no improvements in satisfaction level after receiving the new intervention (which has also been seen in other studies on substitution of care [96,97,98]). However, it may be concluded that an intervention provides added value even if not all Quadruple Aim goals have shown improvement. For example, if costs decrease, patients’ health and HCP work satisfaction increase, but the quality of care experienced by patients remains unchanged, the intervention is of added value. It is essential that all Quadruple Aim goals be assessed and that a balanced conclusion for follow-up be drawn based on these outcomes [99,100]. For included interventions, HCP work satisfaction was found to be improved in the intervention groups, compared with care as usual and with baseline.

For interventions between primary care and secondary or tertiary care (*n* = 6), in the randomized trial designs, no significant differences between intervention and care-as-usual groups were found for most health outcomes. Over time, no improvements were seen in a restricted program containing a three-hour educational intervention [49], whereas improvements were seen in two, more extensive, interventions. These latter comprised more treatment hours, involved more health care disciplines, and consisted of an assessment and, depending on patients’ needs, psychological and exercise treatments. Both of the interventions showing improvement included collaboration with a rehabilitation setting [48,51]. Unfortunately, both also displayed a serious risk of bias and, therefore, more research is warranted before drawing definite conclusions. Regarding cost outcomes, ability to work significantly improved while the return to work after 12 months did not improve in these groups [49,50,52]. Grant et al. found that facilitators such as managing pain, managing work, and making workplace adjustments appear to be key factors for successful return to work [101]. It could be that patients perceived their ability to work sufficiently improved for return to work, but that, for example, workplace adjustments were not yet adequate to allow this. Mixed results were reported regarding quality of care experienced by patients in two different interventions containing an interdisciplinary assessment [52,53], while this was perceived positively in a three-hour educational intervention [49].

The combined intervention in primary care and between primary care and secondary or tertiary care (*n* = 1), described by Stoffer-Marx et al., had some concerns in the risk of bias [54]. The study only showed improvements on the primary health outcome of grip strength, while no difference between groups was seen for pain and health status after two months. Patients receiving the intervention perceived the quality of care as improved after a two-month follow-up, compared to patients receiving care as usual. As this was the only intervention with an extensive interdisciplinary collaboration, no comparisons could be made. However, as future care aims to shift to clinical networks based on collaborations [30], it is important to further explore implementation in clinical practice and research with such collaborations.

An assessment and workplace intervention between primary care and social care (*n* = 2) did not show a significantly greater improvement in health outcomes for the intervention compared with care as usual [55]. Cost outcomes showed a mixture of improved and unchanged results between the intervention and care as usual at different measurement points in a randomized trial design [55], while return to work improved over time after two years in a longitudinal study [56].

In interventions between primary care and secondary or tertiary care and social care (*n* = 2), most health outcomes did not differ between patients who received a workplace intervention combined with graded activity and those who received usual care. For cost outcomes, the duration of sick leave differed between groups in two studies [59,60], but not in another [57]. This could probably be explained by the fact that, in addition to the study of Anema et al. [57], in the studies of Lambeek et al. [59,60], the intervention was extended, involving HCPs of different disciplines (such as a case manager and the patients’ pre-existing specialists), potentially leading to a better effect on duration of sick leave. Due to this interdisciplinary collaboration, such interventions may have a more patient-centered focus and biopsychosocial approach, which could explain the reported results. Currently developed eHealth technologies could make it easier to work interdisciplinarily [102]. However, little research has been performed for both types of workplace interventions. This was also found in the systematic review of Skamagki et al. [103]. In contradiction to our review, they found some consistency in health outcomes for (integrated) workplace interventions. In contrast with our included studies, in the review of Skamagki et al., not all included studies comprised integrated, interdisciplinary interventions [103].

For interventions between primary care and community-based care (*n* = 2), most health outcomes did not differ more in the integrated care condition than with usual care [62,64,66]. No differences in cost outcomes were found between the groups of an intervention existing of combined cognitive behavioral therapy and prescribed exercise compared to treatment as usual (high risk of bias). Quality of care experienced by patients and HCP work satisfaction were found to have both positive and negative results in the interventions. In contrast to our results, other studies found community-based interventions to lower health care costs and improve health outcomes [80,104]. They found that patients often visit HCPs for other complaints than their actual pain or during periods of stress. Easily accessible community-based interventions could take over these kinds of health care visits to both lower costs and increase health.

### 4.1. Strengths and Limitations

This review is the first with an overview of interdisciplinary rehabilitation interventions within primary care and between primary care and other healthcare settings for patients with CMP. The interventions identified cover a broad spectrum of interdisciplinary care interventions with a wide variety of content, duration, and HCP disciplines involved. Moreover, this review is the first focusing on all Quadruple Aim outcomes for integrated interdisciplinary care interventions. Such an overview is valuable given the recommendation of the WHO that rehabilitation services should be integrated within primary care, as well as between primary, secondary, and tertiary levels of health systems, with a case-management role for primary care [16,105]. Another strength of our study is the classification of interventions into subgroups, facilitating comparisons of studies. This classification is based on classifications used earlier and on definitions of intervention types. Moreover, a strength of our study process was the involvement of an information specialist to ensure the quality of the search strategy. Additionally, the PRISMA guidelines for reporting reviews were used. However, a meta-analysis could not be performed as the intervention types and outcome measures were too heterogeneous.

That the selection of articles was limited to those in English, Dutch, or German may have resulted in the exclusion of valid interventions reported in other languages. Unfortunately, interventions were often not described in full detail and/or the health care settings left unclear, potentially resulting in erroneous exclusions of studies. In some studies, it was not clear in which health care setting an HCP, for example, a physician, worked. Furthermore, it was also not always clear how a health care setting should be classified, due to differences between countries and/or the lack of appropriate descriptions. Due to time constraints, it was not possible to contact the authors for additional information. Therefore, potentially relevant interventions (reported with incomplete descriptions of content) may have been excluded in error. In this review, an interdisciplinary care network is defined based on the IASP definition [21]. Thus, all studies with a multimodal treatment provided by a multidisciplinary team (with at least one participating primary care HCP collaborating in assessment and/or treatment using a shared biopsychosocial model and goals) were taken into account. An alternative definition of interdisciplinary care might have led to a different selection of articles. For the randomized trial designs, positive but non-significant results (compared to the control intervention), results with no difference, and results in favor of the control intervention, were all grouped into one category (non-significant (-)). We chose this grouping because not all articles described the results in much detail. A more precise categorization of significant or non-significant results would have given a broader overview.

### 4.2. Implications for Future Innovations and Studies

As future health care shifts to the implementation of clinical networks, more interdisciplinary collaborations will have to be developed and evaluated in the field of rehabilitation for patients with CMP. As it is important that these interventions have a good fit with, and are implemented in, daily health care, we recommend applying co-creation research together with the HCP disciplines involved, patients with CMP, and other stakeholders. In the ideal situation, an evaluation of all Quadruple Aim outcomes needs to be performed with mixed methods to give a full overview of a new interdisciplinary care intervention’s impact.

In order to develop, implement, and evaluate interdisciplinary care interventions across different health care settings, it is recommended that an adjusted version of the IASP definition of interdisciplinary care be used, the current one having been developed for treatments in secondary or tertiary care settings. Moreover, the Quadruple Aim was used to classify the various outcomes in four outcome domains to identify the effect of interventions in these domains. However, in our review, it was difficult to compare the effect of the various interventions on these outcomes, as a wide range of outcome measures and assessment methods were used. Due to the large variation found, a meta-analysis could not be executed. Therefore, to improve uniformity, we propose to develop a core outcome set with measurement instruments and assessment methods with standardized measurement moments for each Quadruple Aim goal (health, quality of care experienced by patients, health care costs, and HCP work satisfaction). This will facilitate future research comparing the effect of interventions. Moreover, not all study designs (e.g., mixed methods or qualitative methods) incorporated Quadruple Aim outcomes so our overview could not be complete. Therefore, it is recommended that interventions developed in the future be evaluated with mixed methods study designs. In this review, a large number of full papers had to be screened before making a decision because most abstracts, and some full papers, did not clearly describe the content of their intervention and the degree of collaboration between HCPs. We recommend that articles use reporting guidelines for abstracts and intervention details, such as the Template for Intervention Description and Replication (TIDieR) checklist [106].

## 5. Conclusions

There is a wide variety in content, collaboration, and evaluation methods of interdisciplinary rehabilitation interventions within primary care, and between primary care and other health care settings, delivering rehabilitation care for patients with CMP. Most interdisciplinary interventions are evaluated in primary care, while fewer interventions are implemented between primary care and other health care settings. It seems that interventions with the involvement of different HCP disciplines, and more patient-centered interventions, with a broader content and duration of treatment, are more effective than care as usual. Therefore, further initiatives and research have to be performed for interdisciplinary care interventions within and across health care settings for patients with CMP. These interventions have to be evaluated with mixed methods on all Quadruple Aim outcomes.

## Figures and Tables

**Figure 1 jcm-10-02041-f001:**
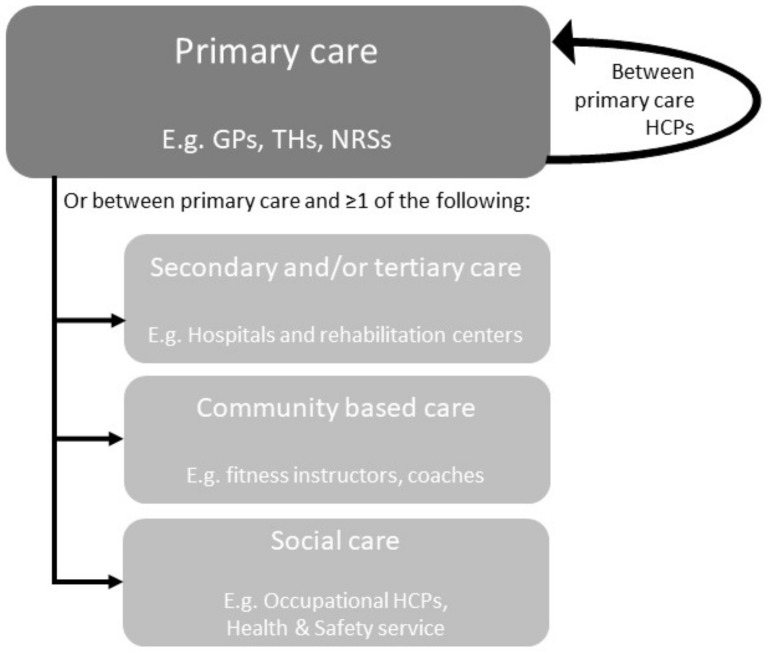
Overview of interdisciplinary care networks. GPs = general practitioners; THs = therapists; NRSs = nurses; HCPs = health care professionals.

**Figure 2 jcm-10-02041-f002:**
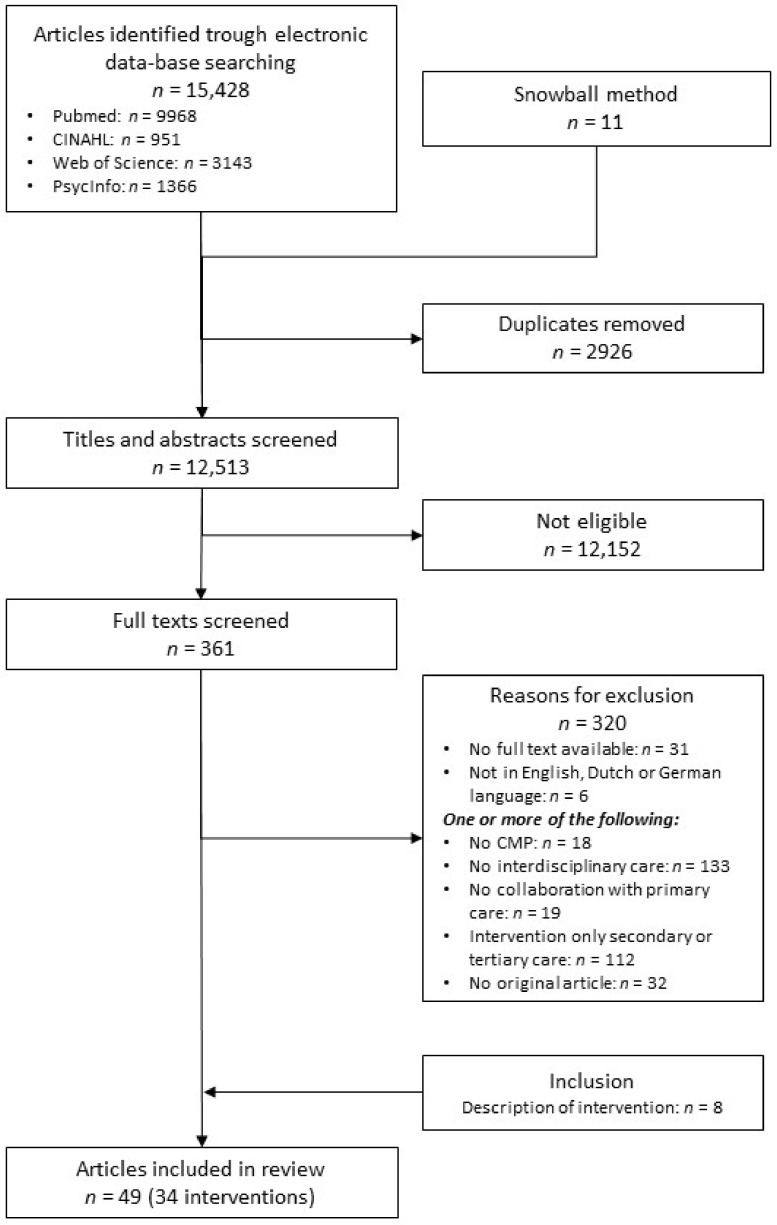
Process of literature selection.

**Figure 3 jcm-10-02041-f003:**
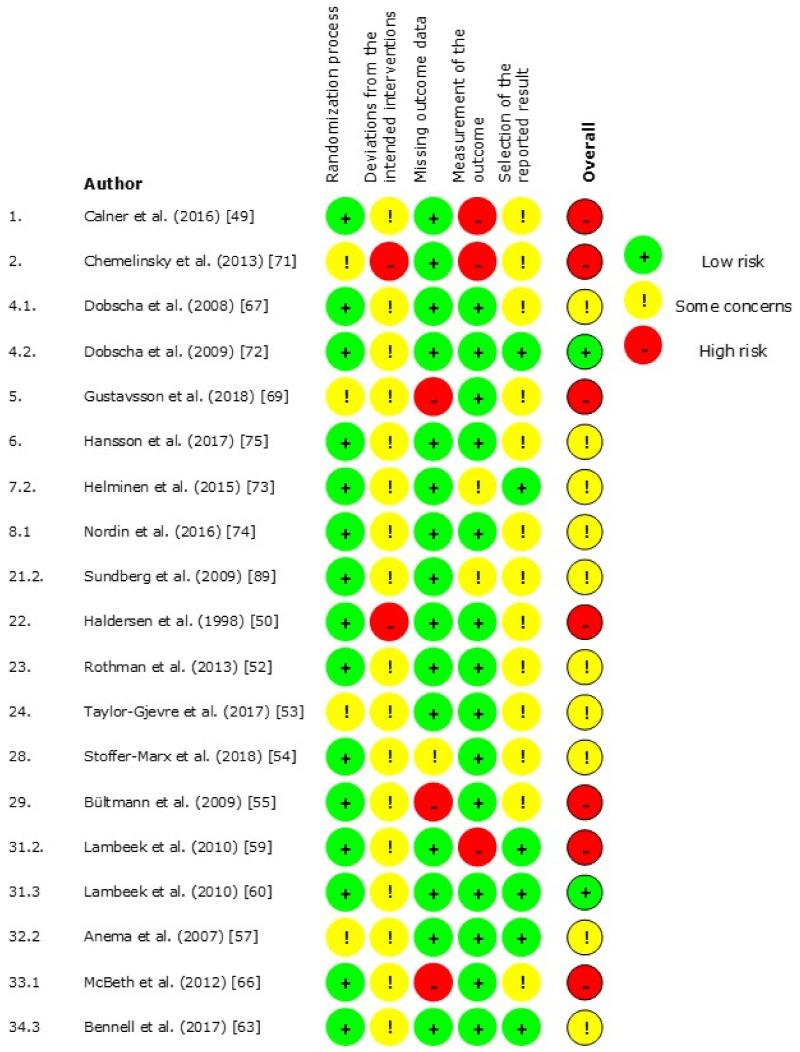
Risk of bias of randomized trial designs.

**Figure 4 jcm-10-02041-f004:**
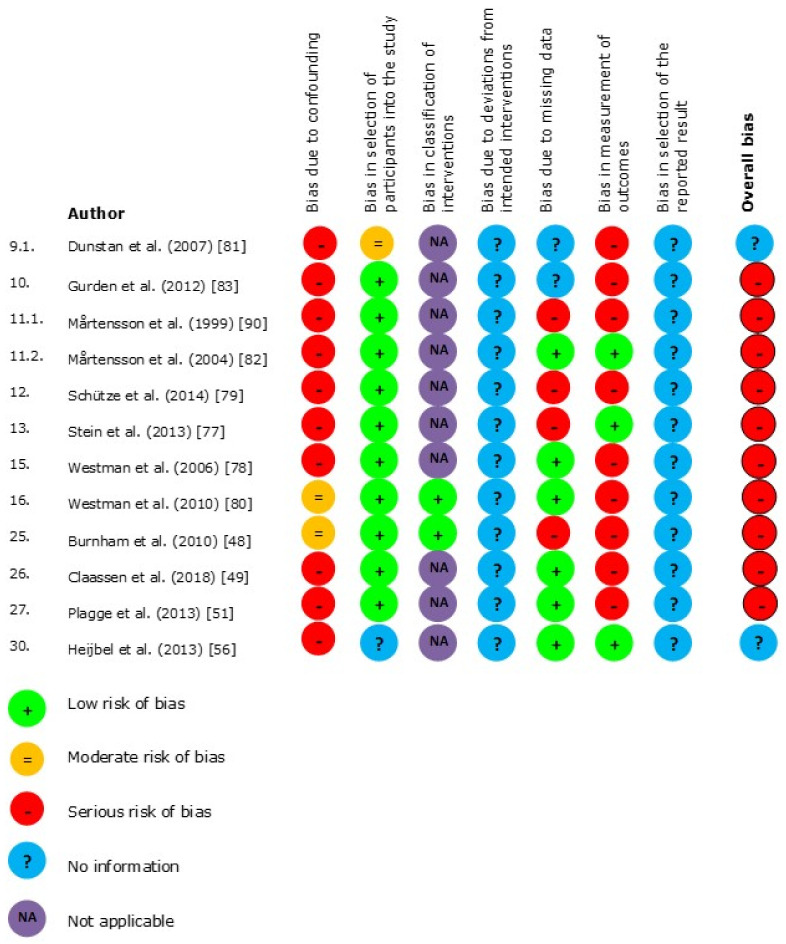
Risk of bias of non-randomized study designs.

**Figure 5 jcm-10-02041-f005:**
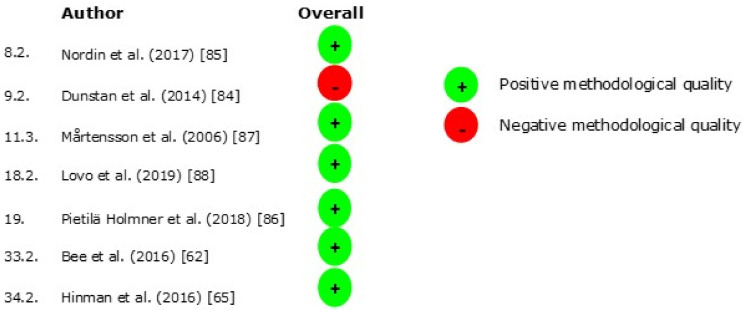
Critical appraisal qualitative designs.

**Table 1 jcm-10-02041-t001:** Inclusion and exclusion criteria.

Inclusion	Exclusion
An intervention for patients with chronic musculoskeletal pain (CMP) of the posture- and locomotion apparatus. Studies were also included if the study population was a mix of patients with subacute and chronic complaints.	An intervention developed for headache or stomach-ache, or only for patients with subacute pain (<12 weeks).
Rehabilitation care enabling individuals aged ≥18 years to maintain or return to their daily life activities, fulfill meaningful life roles and maximize their well-being [30]. The goal of the rehabilitation is on the improvement of participation or functioning of the patient.	A (rehabilitation) intervention which was designed for pre-post surgery care, or if it consisted of eHealth, which substitutes the treatment given by an HCP, or if the intervention only focuses on medication prescription or use.
An interdisciplinary care network based on the IASP definition [21]: *a multimodal treatment provided by a multidisciplinary team collaborating in assessment and/or treatment using a shared biopsychosocial model and goals. The HCPs all have to work closely together with regular team meetings (face to face or online), agreement on the diagnosis, therapeutic aims and plans for treatment and review.* There was a bidirectional discussion or exchange of treatment approaches with the same goal between HCPs of different disciplines (e.g., a GP with a physiotherapist).	An intervention in which HCPs of different disciplines treated a patient but without a mutual goal, bidirectional discussion, or exchange of treatment approaches. An intervention that focuses only on the referral or triage of patients without collaboration during the treatment itself. An intervention with only extended practices roles, e.g., the physiotherapist takes over the roles of the GP.
Implemented within primary care or between primary care and other healthcare settings (secondary or tertiary care, social care, or community-based care) (see Figure 1)	Interventions implemented within or between secondary or tertiary clinic(s).
Original descriptions of (results of) an intervention, such as protocol articles, feasibility studies, process evaluations, and qualitative and quantitative (cost)-effectiveness studies.	A review or guideline. The references for these studies were checked for eligible articles.
Only full texts which were available in Dutch, English, or German.	
Articles published between 1 January 1994 and 14 November 2019.	

**Table 2 jcm-10-02041-t002:** Overview of included studies.

No.	Author, Year & Country	Intervention Name	Target Population	Collaboration	Content and Intensity Intervention
**Within primary care**					
*Randomized trial designs*					
1	Calner et al., (2016) [49] ** Intervention is linked to interventions of Nordin (number 8)* Sweden	Multimodal pain Rehabilitation (MMR) & web behaviour change program for activity (Web-BCPA)	Chronic musculoskeletal pain of the back, neck, shoulders, and/or a generalized pain condition	PH THs PSY or PSY-C NRS 1× Team discussion with patient about treatment plan	**MMR** ≥2×/w, ≥6 w At least 3 different healthcare professionals *Specific Exercise Therapy* *General Exercise Therapy* *Manual Therapy* *Mind-Body Exercise Therapy* **Web-BCPA** 24 h, 7 d, 16 w Self-guided by the patient *Education—Knowledge of disease prevention and ergonomics* *Education—Basic knowledge* *Education—Knowledge of treatment* *Cognitive-behavioral therapy*
2	Chelimsky et al., (2013) [71] US	Primary Practice Physician Program for Chronic Pain (4PCP)	Chronic Pain (back pain 51.9%, fibromyalgia 23.1%, neck pain 6.7%, others)	PH PSY THs	** No separate intervention for patients* Collaborative training of PHs consisting of: **Active learning**: Evidence-based active learning seminars, self-directed learning **Clinical support**: to collaborate with the interdisciplinary treatment team comprising pain-informed THs and PSY providing cognitive-behavioral therapy
3	DeBar et al., (2018) [77] USA	Pain Program for Active Coping and Training (PPACT)	Chronic pain On opioid treatment (≥6 m) On health plan	PPACT interventionist team: PSY-C NCM PCPs: PS PR PPACT interventionists and PCPs meetings for treatment plan (before start treatment) and evaluation (end of treatment)	**Comprehensive intake evaluation** NCM or PSY-C *Assessment* *Medication management* 1 × TS *Education—Knowledge of disease prevention and ergonomics* *Education—Knowledge of treatment* **Cognitive-behavioral therapy (CBT)-based pain coping skills training and adapted movement practice** 12 w (group) *Cognitive-behavioral therapy* **PCP consultation and patient outreach** By NCM and PSY-C
4	1: Dobscha et al., (2008) [67] 2: Dobscha et al., (2009) [72] USA	Study of the Effectiveness of a Collaborative Approach to Pain (SEACAP)	Musculoskeletal pain Chronic Exclusion: fibromyalgia	PSY: care manager IT: intervention & workshop teacher TH: workshop teacher Discussion between PSY and IT about assessment results and treatment recommendations. These are sent by email to clinicians. Leading workshop with PSY and IT or TH.	**Telephone contact** **Written materials** *Education—Basic knowledge* Assessment by PSY *Assessment* *Education—Knowledge of treatment* **Recommendation treatment plan** Based on discussions about symptoms or additional education by PSY and IT **Workshop** 90 m, 4×, 4 months By PSY, co-led by IT or TH *Education—Knowledge of disease prevention and ergonomics* *Education—Knowledge of treatment*
5	Gustavsson et al., (2018) [69] Sweden	Activity and life-role targeting rehabilitation (ALAR)	Musculoskeletal pain Chronic	TSs PH PSY-C TSs: Participating in education meetings about treatment protocol and behavioral medicine approach, 3×,4 h MMR: team discussions about assessment and treatment plan	**Multimodal pain rehabilitation (MMR)** Content and intensity are patient dependent *Assessment* *Cognitive-behavioral therapy* **ALAR + MMR** 1 h, 10×, 10 w Workbook and therapist for goal setting *Assessment* *Education—Knowledge of treatment* *Cognitive-behavioral therapy*
6	Hansson et al., (2010) [75] Sweden	Patient education program for osteoarthritis (PEPOA)	OA in hip, knee or hand	THs OS NRS DT Providing PEPOA	**PEPOA** (*n* = 8–10) 3 h, 5×, 1×/w, 5 w *Education—Knowledge of treatment*
7	1: Helminen et al., (2013) [78] 2: Helminen et al., (2015) [73] Finland	Cognitive-behavioural (CB) intervention for OA	Knee pain Chronic	PSY TH Providing CB intervention	**Cognitive Behavioral group intervention** (*n* = 8–10): 1×/w, 2 h, 6 w *Education—Basic knowledge* *Education—Knowledge of disease prevention and ergonomics* *Education—Knowledge of treatment* *Mind-Body Exercise Therapy*
8	1: Nordin et al., (2016) [74] 2:Nordin et al., (2017) [79] Sweden	Web Behavior Change Program for Activity (Web-BCPA) added to multimodal pain rehabilitation (MMR)	Pain in the back, neck, shoulder, and/or generalized pain	NRS THs PH PSY PSY-C PH for contact with the team and Swedish Social Insurance Agency	**MMR** 2–3×/w, 6–8 w By ≥3 disciplines *Specific Exercise Therapy* *General Exercise Therapy* *Manual Therapy* *Education—Knowledge of disease prevention and ergonomics* *Education—Knowledge of treatment* *Cognitive-behavioral therapy* *Mind-Body Exercise Therapy* **Web-BCPA** 16 w Self-guided *Cognitive-behavioral therapy* *Education—Knowledge of treatment*
*Non-randomized trial designs*					
9	1: Dunstan et al., (2007) [80] 2: Dunstan et al., (2014) [81] Australia	Light multidisciplinary Work-Related Activity Program (WRAP)	Musculoskeletal pain Chronic	PSY TH GP ORP	**WRAP** (*n* = 30, 7 groups) 4 h, 1×/w, 6 w By PSY and TH providing treatment, GP as medical case-manager, and ORP as a return-to-work case manager *Mind-Body Exercise Therapy* *Specific Exercise Therapy* *General Exercise Therapy* *Education—Basic knowledge* *Education—Knowledge of disease prevention and ergonomics* *Workplace intervention*
10	Gurden et al., (2012) [82] UK	North East Essex Primary Care Trust manual therapy service	Back or Neck pain Subacute and chronic	GP CH THs Prescribed treatment plan Advise during referral after treatment (TH to GP)	**GP consultation** *Assessment* *Education—Basic knowledge* *Medication management* **Manual therapy** (within 2 weeks) max. 6× CH, TSs *Manual Therapy* Discharge with a report to GP
11	1: Mårtensson et al., (1999) [83] 2: Mårtensson et al., (2004) [84] 3: Mårtensson et al., (2006) [85] Sweden	Biopsychosocial rehabilitation programme, Focus on Health (FoH)	Pain Chronic	GP NRS THs PSY-C Teaching in FoH program	**FoH** Group sessions (*n* = 5–9) 2×/w, 6 w: 6 × 6 h + 6 × 3 h *Education—Basic knowledge* *Education—Knowledge of disease prevention and ergonomics* *Mind-Body Exercise Therapy* *Education—Knowledge of treatment* Ergonomics Individual introductory and concluding conversation for activity and locomotion analysis
12	Schütze et al., (2014) [86] Australia	Mindfulness-Based Functional Therapy (MBFT)	LBP Chronic	PSY TH Co-facilitated sessions	**MBFT**-group session (*n* = 6 & *n* = 10) 2 h/w, 8 w *Education—Knowledge of treatment* *Mind-Body Exercise Therapy* *Cognitive-behavioral therapy*
13	Stein et al., (2013) [87] Sweden	Multidisciplinary pain rehabilitation (MDR)	Musculoskeletal pain Chronic	GP PSY THs Examination report of GP visible by all Team meeting about biopsychosocial motivation to participate Providing treatment	**MDR** Group sessions (*n* = 6–8): 5 h, 3 d/w, 6 w GP (12 h) *Education—Basic knowledge* *Mind-Body Exercise Therapy* TH (18 h) *Education—Knowledge of disease prevention and ergonomics* *Mind-Body Exercise Therapy* TH (20 h) *Specific Exercise Therapy* *Cognitive-behavioral therapy* *Mind-Body Exercise Therapy* PSY (28 h) *Education—Knowledge of disease prevention and ergonomics* *Education—Knowledge of treatment* *Cognitive-behavioral therapy* Additional education (12 h), provided by Swedish Insurance Agency, Swedish Employment Agency, local fitness center, dietary adviser
14	Tyack et al., (2013) [88] Australia	Student-led interdisciplinary chronic disease health service	Back pain Chronic	NRS PO THs Exercise PSY PSY-C SP DT PR Indigenous health worker Case conference and service delivery	**Intake** by 1 HCP and 2 students *Assessment* **Case conference** By the team Selection of appropriate services **Services** from one or more HCP 3–6 months
15	Westman et al., (2006) [89] Sweden	STAR project; multimodal rehabilitation program	Musculoskeletal pain Chronic Sick listed	PSY PH TH A representative from the National Insurance Company Team discussions about treatment plan	**STAR project** group based (*n* = 8–10) 3.5 h/d, 5 d, 8 w *General Exercise Therapy* *Mind-Body Exercise Therapy* Creative activities *Education—Basic knowledge* *Education—Knowledge of disease prevention and ergonomics* *Education—Knowledge of treatment* **Individual** (when necessary): Physiotherapy or psychotherapy or orthopedic consultation
16	Westman et al., (2010) [90] Sweden	Multidisciplinary rehabilitation program	Musculoskeletal pain Chronic Sick listed	GP TH PSY or PSY-C Team discussions about diagnosis and treatment plan	**Assessment and deciding treatment program** 1×/w By the team *Assessment* And one or more of the following interventions: **Multimodal Group (*n* = 6–8)** 4 h/d, 4 d, 6 w *General Exercise Therapy* *Mind-Body Exercise Therapy* Creative activities *Education—Basic knowledge* *Education—Knowledge of disease prevention and ergonomics* *Education—Knowledge of treatment* **Three-way communication** Patient, GP/PSY or PSY-C Adjustments of treatment plan **Individual** TH or PSY or orthopedic consultation **Workplace-based intervention** *Workplace intervention*
*Qualitative designs*					
17	1:Dorflinger et al., (2014) [91] 2: Purcell et al., (2018) [25] USA	Integrated Pain Team (ITP)	Pain Chronic	PH and/or NP PSY PR Team discussions about diagnosis and treatment plan Providing treatment Keeping track of treatments (inside and outside ITP)	**ITP** existing of: 3×, 2–3 m, *n* = 15–20/m **Interdisciplinary Assessment** 1 h by complete team and patient *Assessment* *Education—Basic knowledge* **Medication management** During complete follow-up by ITP *Medication management* **Additional** *Education—Knowledge of disease prevention and ergonomics* *Education—knowledge of treatment* *Cognitive-behavioral therapy*
18	1: Bath et al., (2016) [65] 2: Lovo et al., (2019) [92] Canada	Secure video conferencing/telehealth	LBP Chronic	TH (urban-based) NP (local rural) 1× Digital assessment	**Digital assessment** 1× NP at patient side performing a physical examination *Assessment* *Education—Knowledge of treatment*
19	Pietilä Holmner et al., (2018) [93] Sweden	Multimodal rehabilitation (MMR)	Pain Chronic Sick listed (or at risk)	THs PH PSY Team discussions about assessment and treatment	**MMR** Individual and/or group intervention *General Exercise Therapy* *Mind-Body Exercise Therapy* *Education—Knowledge of treatment*
20	Stenberg et al., (2016) [94] Sweden	Multimodal rehabilitation (MMR)	Pain Chronic Sick listed (or at risk)	THs PSY-C PSY GP DT NRS THs deliver treatment. Other members deliver treatment or have a consultation function	**MMR** By THs and optionally ≥1 of the other HCPs Group, individually, or combination *Cognitive-behavioral therapy*
21	1: Sundberg et al., (2007) [76] 2: Sundberg et al., (2009) [95] Sweden	Integrative medicine (IM) management	Back or Neck pain Mixed population subacute and chronic	GP Senior CT providers Team discussions about treatment plan	**IM** Conventional therapies, advise by GP *Education—Knowledge of disease prevention and ergonomics* *Anaesthetics* *General Exercise Therapy* Complementary therapies by CT providers 10×, 12 w *Manual Therapy*
**Between primary care and secondary or tertiary care**					
*Randomized trial designs*					
22	Haldorsen et al., (1998) [50] Norway	Multimodal Cognitive Behavioral Treatment (MMCBT)	Back, neck, shoulder pain, generalized muscle pain, more localized musculoskeletal disorders Subacute and chronic Sick listed	NEU GP PSY Registered NRS TH Team discussions on diagnosis and treatment plan Providing treatment (e.g., Education)	**Multidisciplinary rehabilitation** 6 h, 5×/w, 4 w Combination of group and individual treatment *Assessment* *Specific Exercise Therapy* *General Exercise Therapy* Individual by TH *Cognitive-behavioral therapy* (8×) *Education—Basic knowledge* *Education—Knowledge of disease prevention and ergonomics* *Education—Knowledge of treatment* 2× lectures and discussions by all healthcare professionals *Workplace interventions* By physician, human resource officer, occupational counsellor, representative of a governmental social insurance authority.
23	Rothman et al., (2013) [52] Sweden	Multidisciplinary, multimodal (MM), multi-professional assessment	CMP Chronic	GP And ≥3: NRS PSY TH PSY-C OS when necessary: liaison PH at the Psychosomatic Medicine Clinic (PMC) Interdisciplinary team meeting about assessment	**Assessment in the MM Group** Each discipline had 1 meeting with patient (mean 7 sessions) Conference meeting to give treatment advice: Multidisciplinary group pain management at the PMC, Multidisciplinary individual pain management at the PMC, Multidisciplinary individual pain management at GP and associated team or at a multidisciplinary clinic. *Assessment*
24	Taylor-Gjevre et al., (2017) [53] Canada	Video-conferencing	RA	Urban-based RT On-site TH Performing assessment and follow-up care	**Video-conferencing treatment** 4× TH is at patient side for physical examination and set up conferencing with the rheumatologist who is performing the assessment and follow-up care *Assessment*
*Non-randomized trial designs*					
25	Burnham et al., (2010) [48] Canada	Central Alberta Pain and Rehabilitation Institute (CAPRI) program	Pain Chronic	PH TH GP PSY NRS DT KN Team discussions about treatment plan Executing of treatment (full multidisciplinary management)	**Referral documentation review** GP **Initial assessment** 1: spine care assessment: 1,5 h, by PH and TH 2: medical care assessment (optional): 2 h, by GP *Assessment* **Treatment (1 of the options)** I-1: Consultation only: education, activity modification, and a customized home exercise program *Education* *General exercise therapy* I-2: Interventional management: anaesthetic block by PH *Anaesthetics* I-3: Supervised medication management: by GP *Medication management* I-4: Full multidisciplinary management (*n* = 4–6): 5 h, 1×/w, 2–3 months, the whole team Group discussions about education and treatment plan *Psychotherapy* *Education—Basic knowledge* *Education—Knowledge of treatment*
26	Claassen et al., (2018) [49] The Netherlands	Osteoarthritis (OA) education	OA in hip or knee	GP TH OS or NP Public health advisor (when available) Teaching in OA educational program	**OA educational program** (*n* = 10–12) 1,5 h, 2× *Education—Knowledge of disease prevention and ergonomics* *Education—Basic knowledge* *Education—Knowledge of treatment* **Booklet** Information, monitoring forms, course handout, 20 FAQs, a pedometer, and a list of websites and contact information
27	Plagge et al., (2013) [51] USA	Integrated Management of Pain and PTSD in Returning OEF/OIF/ONDVEterans (IMPPROVE)	Pain Chronic Posttraumatic stress disorder Veterans	PSY PH Discussions about assessment and weekly telephone meetings about treatment	**Biopsychosocial evaluation** 90 min by PSY *Assessment* **Care management** 1×/w by PSY and PH Reviewing recommendations with veterans, assessing interest and willingness to engage in recommended treatments, discussing concerns or questions, coordination of care between services, facilitating communication between the veteran and providers, helping veterans navigate the VA system, monitoring treatment plans **Behavioral Activation Psychotherapy** 8×, 75–90 min Individual by PSY *Cognitive-behavioral therapy*
**In primary care and between primary care and secondary or tertiary care**					
*Randomized trial design*					
28	Stoffer-Marx et al., (2018) [54] Austria	The combined intervention	OA in hand Chronic	RT THs NRS DT They have primary or specialized care setting expertise (or both) Two deliver treatment together	**Baseline assessment** By blinded assessor *Assessment* **The combined intervention** Individual treatment 7×/w, 8 w By 2 HCPs *Specific Exercise Therapy* *Medication management* *Education—Knowledge of disease prevention and ergonomics* *Education—Knowledge of treatment*
**Between primary care and social care**					
*Randomized trial designs*					
29	Bültmann et al., (2009) [55] Denmark	Coordinated and Tailored Work Rehabilitation (CTWR)	Musculoskeletal disorders or LBP Subacute and chronic Sick listed	OP TH CH PSY PSY-C Team discussions about diagnosis and treatment plan PSY-C as caseworker establishing and maintaining contact with the workplace and the municipal case manager Report of a work rehabilitation plan to GP	**CTWR**: existing of **Work disability screening** 1×, 2 h, 4–12 w after sick leave Interdisciplinary *Assessment* **Work rehabilitation plan** max. 3 months Interdisciplinary with patient *Assessment* *Workplace intervention*
*Non-randomized trial design*					
30	Heijbel et al., (2013) [56] Sweden	Occupational Health Service (OHS)	Mixed group Musculoskeletal problems Subacute and chronic Sick listed	PH TH PSY NRS Team assessment	**Team assessment** With team *Assessment* **Rehabilitation meeting and plan of measures** With patient, supervisor, several OHS team members, local insurance office, trade union (optional) **Treatment** *Education* *Cognitive-behavioral therapy* 4 w, FU 6 m or 12 m *Workplace intervention* **Follow-up meeting after rehabilitation**
**Between primary care and secondary or tertiary care and social care**					
*Randomized trial designs*					
31	1: Lambeek et al., (2007) [58] 2: Lambeek et al., (2010) [59] 3: Lambeek et al., (2010) [60] **Interventions are identical to the interventions of Steenstra and Anema. (number 32)* The Netherlands	Multidisciplinary outpatient care program (MOC)	Non-specific LBP Chronic Sick listed <2 years	CM: coordination of care and communication team (primary-tertiary care) THs Patients own medical specialist GP OP Conference call with team: 1×/3 w	**MOC** existing of: **Case management protocol** CM collect information from HCP team. Referral in collaboration with OP. Organization of conference calls. *Assessment* **Workplace intervention protocol** 8 h, 4 w TH helps to achieve consensus between patient and supervisor for return to work *Workplace intervention* **Graded activity program** max. 26 sessions, max. 12 weeks Local TH practices *Cognitive-behavioral therapy*
32	1: Steenstra et al., (2003) [61] 2: Anema et al., (2007) [57] **Interventions are identical to the interventions of Lambeek. (number 31)* The Netherlands	Workplace intervention and Graded Activity	Non-specific LBP Subacute and chronic Sick listed	1: OP GP Contact about referral 2: OP GP THs Workplace intervention with worker, employer, OP, GP	**Combined intervention (CI)**: existing of **Workplace intervention (WI)** direct after inclusion (2–6 weeks after sick leave), 24 d *Assessment* *Workplace intervention* **Graded Activity Program (optional)** 0.5 h, 2×/w, max. 26× After 8 weeks of sick leave By TH *Cognitive-behavioral therapy*
**Between primary care and community-based care**					
*Randomized trial designs and qualitative designs*					
33	1: McBeth et al., (2012) [66] 2: Bee et al., (2016) [62] England	Combined Cognitive Behavioral Therapy (T-CBT) and prescribed exercise (PE)	Fibromyalgia Chronic	TH FI Two-way information exchange between TH and FI	**T-CBT** 1 h telephone assessment 30–45 min, 1×/w, 7 w, FU 3 and 6 m By TH *Education—Knowledge of treatment* *Cognitive-behavioral therapy* **PE** 20–60 h-2×/w By FI *General exercise therapy*
34	1: Bennell et al., (2012) [64] 2: Hinman et al., (2015) [65] 3: Bennell et al., (2017) [63] Australia	Physiotherapy plus telephone coaching	Patients with knee OA Subacute and chronic	TH TC Written information exchange between the TC and TH occurred after each session.	**Physical therapy program** 30 m, 5×, 6 months *Specific exercise therapy* *General exercise therapy* **Information booklet** *Education—Knowledge of disease prevention and ergonomics* **Telephone coaching** 6–12×, 6 months *Cognitive-behavioral therapy*

Care manager (CM), Chiropractor (CH), Dietician (DT), Fitness instructors (FI), General practitioner/Primary care physician (GP), Internist (IT), Kinesiologist (KN), Neurologists (NEU), Nurse (NRS), Nurse care managers (NCM), nurse practitioners (NP), Occupational health nurse (OHN), Occupational physician (OP), Occupational rehabilitation providers (ORP), Orthopaedic surgeon/specialist (OS), Pharmacists (PR), Physician (Physiatrist, Rehabilitation physician) (PH), Podiatrists (PO), Psychologist (PSY), Psychosocial counsellor (Behavioural specialist, Counsellors, psychotherapist, Social worker) (PSY-C), Rheumatologist (RT), Speech pathologist (SP), Telephone coaches (TC), Therapists (Ergonomist, Occupational physiotherapist, Occupational therapist, Osteopaths, Physiotherapists) (TH). Low back pain (LBP), chronic musculoskeletal pain (CMP), osteoarthritis (OA), rheumatoid arthritis (RA).

**Table 3 jcm-10-02041-t003:** Overview of study designs, study outcomes, and results based on the Quadruple Aim.

	Author & Year	Study Date	Study Design & N	Study Outcomes	Results
**In primary care**					
*Randomized trial designs*					
1	Calner et al.,	2011–2014	RCT	*Health*	
	(2016) [49]			* Pain intensity (100-mm Visual Analogue Scale)	4 m:-1 y: **-**
			*n* = I:60, C:49	* Pain-related disability (Pain Disability Index)	4 m:-1 y: **-**
				* Health-related quality of life (36-item Short-Form Health Survey)	*All domains*: 4 m:-1 y: **-**
				*Costs*	
				* **Work-related aspects and behavior §** (Work Ability Index (7–49))	4 m:-1 y: **-**
				* **Working percentage §**	+/-
2	Chelimsky et	-	Controlled pilot	*Health*	*n.d between groups, over time:*
	al., (2013) [71]		study	* Pain intensity (0–10 Numeric Rating Scale)	0 m-1 y: +
				* Pain qualities (Short-Form McGill Pain Questionnaire)	0 m-1 y: +
			*n* = 40 pt	* Physical functioning; measured with:	
			*n* = I:12, C:16	- Multidimensional Pain Inventory Interference Scale	0 m-1 y: +
			HCPs	- Brief Pain Inventory	0 m-1 y: +
				- Multidimensional Health Locus of Control Scale	0 m-1 y: -
			HCPs are	* Emotional functioning; measured with:	
			controlled, not	- Back Depression Inventory	0 m-1 y: +
			the pts	- Profile of Mood States	0 m-1 y: +
				*Experienced quality of care by patients*	
				* Participant ratings of global improvement and satisfaction with treatment;	
				measured with:	
				- Patient Global Impression of Change	*n.d.*
				- Treatment helpfulness questionnaire	*n.d.*
				- Facilitation of patient involvement in care	+
				*Satisfaction with work by HCPs*	*n.d between groups, over time:*
				* **Experiences with work ¥** (24-item physician perspectives questionnaire)	
				- Knowledge	I: 0 m-1 y:- C: 0 m-1 y: -
				- Diagnosis/Management	I: 0 m-1 y: + C: 0 m-1 y: +
				- Treatment Comfort	I: 0 m-1 y:- C: 0 m-1 y: -
				- Treatment Satisfaction	I: 0 m-1 y: + C: 0 m-1 y: -
				- Use of Referrals	I: 0 m-1 y: + C: 0 m-1 y: -
				* Interview regarding: MD functional approach, Patient functional approach,	*All:* +
				Enabling self-management, Assessing patient mood, Assessing patient sleep,	
				Comfort with use of medication	
3	DeBar et al.,	2014–2017	Protocol	*Health*	*n.a.*
	(2018) [77]		Randomized	* **Pain, Enjoyment, General Activity (PEG) §** (3-item measure based on Short Form	
			pragmatic trial	of the Brief Pain Inventory)	
				* Pain-related disability (Roland Morris Disability Questionnaire)	
			Intended *n* = 851		
			pt in clusters	*Costs*	
				* Healthcare utilization (opioids dispensed, both aggregated and disaggregated	
				primary care contact, use of specialty pain services, inpatient services related to	
				pain, and overall outpatient utilization)	
				*Experienced quality of care by patients*	
				* Patients’ satisfaction with their primary care services (one question)	
				* Satisfaction with overall pain-related services provided by the health plan (one	
				question)	
4	1: Dobscha et	2006–2007	1: RCT (baseline	1: *Health*	*n.a. (only baseline results)*
	al., (2008) [67]		findings)	* Quality of life (EuroQoL-5D)	
				* **Pain-related function/disability §** (Roland Morris Disability Questionnaire)	
			I: 187 pt,	* Pain severity (Chronic Pain Grade Severity subscale)	
			20 HCPs	* Depression severity (Patient Health Questionnaire)	
			C: 214 pt,	* Comorbidity (Chronic Disease Score (RxRisk-V [pharmacy data]))	
			22 HCPs	* Readiness for change (modeled after Epler)	
				* Global Impression of Change	
				*Costs*	
				* Opioid prescriptions (number, type, doses, duration)	
				* Use of adjuvant pain medications	
				* Concurrent use of multiple short-acting opioids	
				* Utilization and costs (primary care, pain specialty, mental health/SUD specialty,	
				emergency, other ambulatory treatment visits, contact, inpatient days)	
				*Experienced quality of care*	
				* Global Care Satisfaction	
				* Survey of Health Experiences of Veterans (pain care, 1-item)	
				*Work satisfaction by HCPs*	
				* Pain management attitudes/behaviors items	
				* Job satisfaction	
				* Provider helpfulness of intervention	
	2: Dobscha et	2006–2007	2: RCT	2: *Health*	
	al., (2009) [72]		*n* = I:187, C:214	* Quality of life (EQ-5D)	0–1 y: -
				* **Pain-related function/disability §** (Roland Morris Disability Questionnaire)	0–1 y: +
				* Pain intensity (Chronic Pain Grade Severity subscale)	0–1 y: +
				* Depression severity (Patient Health Questionnaire)	0–1 y: +
				* Global Impression of Change	0–1 y: +
				*Costs*	
				* Opioid prescriptions (number, type, doses, duration)	0–1 y: +
				* Use of adjuvant pain medications, use of multiple short-acting opioids	0–1 y: -
				* Utilization and costs (Primary care, pain specialty, mental health/SUD specialty,	0–1 y: +/-
				emergency, other ambulatory treatment visit, and contacts; inpatient days)	
				*Experienced quality of care by patients*	
				* Global Care Satisfaction	0–1 y: -
5	Gustavsson et	2011–2013	Feasibility study	*Health*	*n.a. (only baseline results)*
	al., (2018) [69]		Pragmatic RCT	* Health-related quality of life (EuroQoL-5D)	
				* Disability	
			*n* = I:15, C:17 pt	* Pain intensity	
			*n* = 7 HCPs	* Pain catastrophizing	
				* Pain-related fear-avoidance	
				* Depression	
				* Anxiety	
				*Costs*	
				* Sickness absence	1 y: -
				* Costs-utility	9 w: +/- 1 y: +
				*Experienced quality of care by patients*	
				* Patients’ satisfaction with treatment (Self-assessment questionnaire)	9 w:-1 y: -
				*Satisfaction with work by HCPs*	
				* **Perceived usability of the program ¥** (interview)	+
				* **Proficiency in applying the techniques and delivering of the intervention**	+
				**components ¥** (interview)	
6	Hansson et al.,	-	RCT	*Health*	
	(2010) [75]			* **Self-perceived health ¢** (EuroQol-5D)	6 m: Index:-VAS: +
			*n* = I:61, C:53	* Function lower extremities (one-leg rising from sitting to standing)	6 m: -
				* Balance performance; measured with:	
				- standing one leg eyes open	6 m: -
				- standing one leg eyes closed	6 m: +
				* Function upper extremities (Grip Ability Test)	6 m: -
7	1: Helminen	2011–2012	1: Protocol RCT	*1: Health*	*n.a.*
	et al., (2013)			* **Self-reported pain §** (pain subscale of the Western Ontario and McMaster	
	[78]		Intended *n* =	Universities Osteoarthritis Index)	
			I:54, C:54	* Physical functioning and stiffness (corresponding subscales of the Western Ontario	
				and McMaster Universities Osteoarthritis Index)	
				* Pain intensity (0–10 Numeric Rating Scale)	
				* Health-related quality of life (RAND-36 item Health Survey and 15-dimensional	
				Health-related Quality of Life)	
				* Life satisfaction (4-item Life Satisfaction)	
				* Kinesiophobia (Tampa Scale for Kinesiophobia)	
				* Catastrophizing (Pain Catastrophizing Scale)	
				* Depressive symptoms (Beck Depression Inventory)	
				* Global assessment of change	
				*Costs*	
				* Use of analgesics, topical pain medication (patient reports)	
				* Number of intra-articular injections	
				* Use of health services	
				* Number of sick-leave days	
				* Cost-effectiveness (QALY)	
	2: Helminen	2011–2012	2: RCT	*2: Health*	
	et al., (2015)			* **Self-reported pain §** (pain subscale of the Finnish version of the Western Ontario	3 m: -
	[73]		*n* = I:55, C:56	and McMaster Universities Osteoarthritis Index)	
				* Physical functioning and stiffness (corresponding subscales of the Western Ontario	3 m: -
				and McMaster Universities Osteoarthritis Index)	
				* Pain intensity (0–10 Numeric Rating Scale)	3 m: -
				* Health related quality of life (RAND-36 item Health Survey and 15-dimensional	3 m: -
				Health-related Quality of Life)	
				* Life satisfaction (4-item Life Satisfaction)	3 m: -
				* Kinesiophobia (Tampa Scale for Kinesiophobia)	3 m: -
				* Catastrophizing (Pain Catastrophizing Scale)	3 m: -
				* Depressive symptoms (Beck Depression Inventory)	3 m: -
				* Global assessment of change	3 m: -
				* BMI (weight/length2)	3 m: -
				*Costs*	
				* Pain medication	3 m: -
				* Use of health services	3 m: -
				* Number of sick-leave days	3 m: -
8	1: Nordin et	2011–2015	1: RCT	*1: Health*	
	al., (2016) [74]			* **Pain intensity ¶** (100-mm Visual Analogue Scale)	4 m:-1 y: -
			*n* = I:55, C:43		
				*Experienced quality of care by patients*	4 m & 1 y: *intervention:* +
				* Patients’ satisfaction with the intervention (2-items)	4 m & 1 y: *own effort:* -
				*Costs*	*no significance calculated*
				* Intervention characteristics	
				* Health care consumption	
				* Sick leave	
	2: Nordin et	2011–2015	2: Qualitative	*2: **Experienced quality of care by patients ¶***	
	al., (2017) [79]		interviews	* Experiences of patient participation in the rehabilitation and intervention	
				Theme: It’s about me	
			*n* = 19	- Take part in a flexible framework of own priority	+/-
				- Acquire knowledge and insights	+
				- Ways toward change	+/-
				- Personal and environmental conditions influencing participation	+/-
*Non-randomized trial designs*					
9	1: Dunstan et	-	1: Pilot study	1: *Health*	*Pre- post program:*
	al., (2007) [80]		Uncontrolled	* Pain severity (0–10 Numeric Rating Scale)	+
			repeated	* Mood (Depression, Anxiety, Stress Scales)	+
			measures design	* Disability (Modified Roland Morris Disability Questionnaire)	-
				* Catastrophizing (Pain Catastrophizing Scale)	+
			*n* = 30	* Fear-avoidance (Tampa Scale for Kinesiophobia)	+
				*Costs*	
				* **Paid work participation ¥** (for any number of hours)	6 m: -
	2: Dunstan et	-	2: Qualitative	2: ***Experienced quality of care by patients ¶***	
	al., (2014) [81]		design	* How much the program helped them to manage their pain, become more active,	+/-
				and get back to work (5-point Likert-type scales)	
			*n* = 33	* The helpfulness of each component of the program (5-point Likert-type scales)	+/-
				* The quality of the psychologist’s and physiotherapists’ input (5-point Likert-type	+
				scales)	
				* Program improvements	*n.d.*
10	Gurden et al.,	2009–2010	Uncontrolled	*Health*	*Baseline and discharge:*
	(2012) [82]		pilot study	* **Back and neck pain ¥** (Bournemouth Questionnaire)	*+*

			*n* = 696	*Costs*	
				* Medication usage	*+*
				* Other healthcare utilization	*+*
				* Work status	*no results described*
				*Experienced quality of care by patients*	
				* a patient satisfaction with treatment scale (5-point scale)	*+*
11	1: Mårtensson	2004	1: Longitudinal	1: *Health*	
	et al., (1999)		pre-post test	* **General well-being ¥** (100-mm Visual Analogue Scale)	0 m-2 m: + 0 m- 2 y: +
	[83]		design	* Pain management ability (100-mm Visual Analogue Scale)	0 m-2 m:- 0 m- 2 y: + 2 m-2 y: +
				* Perceived complaints (100-mm Visual Analogue Scale)	0 m-2 m: + 0 m- 2 y: +
			*n* = 70	* Influence of the intervention and perceived change due to treatment (Personality-	*Body awareness: + Other: -*
				Physical-Cognitive)	
	2: Mårtensson	2002–2003	2: A longitudinal	2: *Costs*	
	et al., (2004)		intervention	* **Sick leave days ¶** (Statistics register at the social insurance office)	0 m-1 y:- 0 m-2 y: + 1 y-2 y: +
	[84]		study design	* Doctor visits (Statistics register at the county council in question)	0 m-1 y: + 0 m-2 y: + 1 y-2 y: +
				* Level of absenteeism due to occupational disability (Statistics register at the social	0 m-post: + 0 m-1 y: + 0 m-2 y: +
			*n* = 54	insurance office)	
	3: Mårtensson	2002–2003	3: Explorative	3: ***Experienced quality of care by patients §***	
	et al., (2006)		descriptive	* Content, format, the group’s role, the leader’s role, and the participant’s role	
	[85]		qualitative	- A place to which you belong	*+*
			design	- An encouraging environment	*+*
				- Expectations of being regarded as a sick person	*-*
			*n* = 24	- The value of one’s own contribution	*+*
				- Reacting but not acting	*-*
				- Awareness and integration	*-*
12	Schütze et al.,	-	Pilot study	*Health*	
	(2014) [86]		Repeated	* Risk of future disability (Örebro Musculoskeletal Pain Questionnaire)	0 m-3 m:-0 m-6 m: -
			measures design	* Low-back related functional disability (Oswestry Disability Questionnaire)	0 m-3 m: + 0 m-6 m: -
				* Emotional functioning (Short form of the Depression, Anxiety, Stress Scales)	
			*n* = 12	- Depression	0 m-3 m:-0 m-6 m: +
				- Anxiety	0 m-3 m:-0 m-6 m: -
				- Stress	0 m-3 m: + 0 m-6 m: +
				* Present-moment awareness of actions, interpersonal communication, thought,	0 m-3 m:-0 m-6 m: +
				emotions, and physical state (Mindful Attention Awareness Scale)	
				* Catastrophizing (Pain Catastrophizing Scale)	0 m-3 m: + 0 m-6 m: +
				* **Health status and health-related quality of life ¶** (36-item Short-Form Health	*All:* 0 m-3 m: + 0 m-6 m: +
				Survey)	
				*Experienced quality of care by patients*	
				* Patient satisfaction (Client Satisfaction Questionnaire)	*+*
13	Stein et al.,	2008–2011	Controlled	*Health*	
	(2013) [87]		pragmatic trial	* Pain intensity (0–10 Numeric Rating Scale)	1 y: -
				* Anxiety and depression (Hospital and Anxiety Depression Scale)	
			*n* = 59	- Anxiety	1 y: -
				- Depression	1 y: +
				* Pain severity (Multidimensional Pain Inventory)	1 y: -
				* Health-related quality of life; measured with:	
				-36-item Short-Form Health Survey, social function	1 y: +
				- EuroQoL-5D, physical function	1 y: -
				*Costs*	
				* Sick-leave (Software-system “Swedestar”)	1 y: +
				* **Consumption of opioids §** (Software-system “Swedestar”)	1 y: -
				* Healthcare utilization	1 y: +
14	Tyack et al.,	2010	Protocol	*Health*	*n.a.*
	(2013) [88]		Longitudinal	* **Health § (36-item Short-Form Health Survey)**	
			cohort study	* BMI (weight/length2)	
				* Waist circumference	
			Intended *n* = 130	* Psychological distress (6-item Kessler)	
				* Disease burden (self-report comorbidity measure)	
				* Comorbid conditions (self-report comorbidity measure)	
				* Perceived functional and structural social support (Medical Outcomes Study Social	
				Support Survey)	
				* Self-reported health perception (one question)	
				*Costs*	
				* Hospital utilization (number of days spent in hospital, and number of hospital	
				admissions)	
				* Healthcare costs (Medicare and hospital utilization records)	
15	Westman et	1994–1996	Cohort	*Health*	
	al., (2006) [89]			* **Quality of life-life satisfaction ¥** (10-items self-constructed questionnaire)	0 m-1 y:-0 m-5 y: +
			*n* = 72	* Intensity of pain and frequency (100-mm Visual Analogue Scale)	0 m-1 y: + 0 m-5 y: +
				* Function (Disability Rating Index)	0 m-1 y:-0 m-5 y: +
				* Anxiety and depression (Hospital and Anxiety Depression Scale)	0 m-1 y:-0 m-5 y: +
				* Health profile assessment	0 m-1 y: + 0 m-5 y: +/-
				*Experienced quality of life by patients*	
				* Patient satisfaction (three questions)	+
				*Costs*	
				* Sick leave/Return to work (self-reported data)	+/-
				* Job strain (11-items self-constructed questionnaire)	0 m-1 y: + 0 m-5 y: -
16	Westman et	1998–2000	Trial with	*Health*	
	al., (2010) [90]		control group	* **Health-related quality of life ¥** (36-item Short-Form Health Survey)	3 y: -
				* Coping (Coping strategies questionnaire)	3 y: -
			*n* = I: 59, C:52	* Catastrophizing (Pain Catastrophizing Scale)	3 y: -
				* Fear of movement (Tampa Scale for Kinesiophobia)	3 y: -
				* Psychosomatic symptoms	3 y: -
				*Costs*	
				* Work capacity/sick leave (reported by patients)	3 y: -
				* Job strain (11-items, self-constructed questionnaire)	3 y: -
				* Health care utilization (how many visits (0 to 10) during the past 12 months)	
				- GP	3 y: +
				- Physiotherapist	3 y: -
				- Naprapath or chiropractor	3 y: +
				* Drug consumption (one question)	3 y: -
*Qualitative designs*					
17	1: Dorflinger	2015–2016	1: Description of	1: n.a.	*n.a.*
	et al., (2014)		intervention		
	[91]				
	2: Purcell et	2015–2016	2: Mixed	2: *Work satisfaction by HCPs*	*n.a. mixed population*
	al., (2018) [25]		methods trial	* **Perspectives on the perceived effectiveness of the chronic pain care provided to**	
				**their patients ¥**	
			Interview: *n* = 61	* Job satisfaction, stress level, and burnout	
			Questionnaire:	* Confidence in and comfort with providing chronic pain care	
			*n = * 65	* Provider confidence in and satisfaction with chronic pain care and the intervention	
				(questionnaire)	
				* Burnout measure (1 item, based on Maslach Burnout Inventory)	
18	1: Bath et al.,	2014–2015	1: Protocol RCT	1: *Health*	*n.a.*
	(2016) [65]			***Self-perceived function §** (Modified Oswestry Disability Questionnaire)	
			Intended *n* =	* Pain intensity (0–10 Numeric Rating Scale)	
			I:20 C:20	* Quality of life/general health status (EuroQoL-5D)	
				*Costs*	
				* Costs (self-report diaries: intervention/treatment costs, work status, absenteeism	
				and disability days related to back pain, health service use, other pain-related costs,	
				and costs from participation in the study)	
				*Experienced quality of care by patients*	
				* Patient satisfaction (a modified version of the Visit Specific Satisfaction Instrument)	
				*Work satisfaction by HCPs*	
				* Satisfaction of HCPs with intervention (semi structured interview)	
	2: Lovo et al.,	2014–2015	2: Qualitative	2: *Experienced quality of care by patients*	
	(2019) [92]		design	* Patient satisfaction (a modified version of the Visit Specific Satisfaction Instrument,	*+*
				a space for comment, and semi-structured interviews)	
			Questionnaire:		
			*n* = 19 pt	*Work satisfaction by HCPs*	
			Interview:	* **Satisfaction of HCPs with PT-delivered telehealth assessments §**	
			*n* = 2 HCP,	(semi structured interview)	
			*n* = 6 pt	- Access to care	*+*
				- Effective interprofessional practice	*+*
				- Enhanced clinical care	*+*
				- Technology	*+/-*
19	Pietilä	-	Qualitative	*Experienced quality of care by patients*	
	Holmner et al.,		interviews	* **Experiences of MMR §** (interview)	
	(2018) [93]			- from discredited towards obtaining redress	*+/-*
			*n* = 12	- from uncertainty towards knowledge	*+*
				- from loneliness towards togetherness	*+/-*
				- “acceptance of pain”: an ongoing process	*+/-*
20	Stenberg et al.,	2013–2014	Qualitative	***Satisfaction with work by HCPs §***	*n.a. mixed population*
	(2016) [94]		design	* Benefits and drawbacks of including patients in MMR	
				* Types of patients in MMR	
			*n = * 14	* Factors that facilitate or impede conduct of MMR	
				* Professional views on MMR	
				* Teamwork experiences	
				- Select patients for success	
				- Multilevel challenge	
				- Ethical dilemmas	
				- Considering what is a good result	
21	1: Sundberg et	2003–2006	1: Qualitative	1: Barriers and facilitators: Notes of research group meetings (*n* = 40) and	*n.a.*
	al., (2007) [76]		study design	field notes from seminars and lectures about results from the development	
				and implementation phases	
				*Outcomes are not one of the Quadruple Aim outcomes*	
	2: Sundberg et		2: Feasibility	2: *Health*	
	al., (2009) [95]		study	* **Health-related quality of care §** (36-item Short-Form Health Survey)	16 w: *All subscales: -*
			Pragmatic RCT	* Disability (0–10 Numeric Rating Scale)	16 w: -
				* Stress (0–10 Numeric Rating Scale)	16 w: -
			*n* = I:36, C:27	* Well-being (0–10 Numeric Rating Scale)	16 w: -
				* Days in pain (0–10 Numeric Rating Scale)	16 w: -
				*Costs*	
				* Use of analgesics (0–10 Numeric Rating Scale)	16 w: -
				* Use of health care (0–10 Numeric Rating Scale)	16 w: -
**Between primary care and secondary or tertiary care**					
*Randomized trial designs*					
22	Haldorsen et	-	RCT	*Health*	
	al., (1998) [50]			* Quality of life (6-items, self-constructed questionnaire)	1 y: +/-
			*n* = I:312; C:157	* Pain intensity (drawing test and 100-mm Visual Analogue Scale)	*n.d.*
				* Amount of pain caused by daily activities (Activity Discomfort Scale)	1 y: +/-
				* Subjective health (Ursin’s Health Inventory)	1 y: +
				* Anxiety (Spielberger State Trait anxiety Scale)	1 y: -
				* Psychological distress (brief version of the Hopkins Symptom Check List)	1 y: +
				* Health locus of control (Multidimensional Health Locus of Control Scale)	1 y: +
				* Physiotherapy examination (functional ability, movement, relaxation ability, pain,	1 y: +/-
				aerobic capacity test, practical skills)	
				*Costs*	
				* **Return to work after 12 months §**	1 y: -
				* Subjective work ability (Graded Reduced Work Ability scale)	1 y: +
23	Rothman et	2001–2004	RCT	*Health*	
	al., (2013) [52]			* **Pain intensity ¢** (100-mm Visual Analogue Scale)	15 m: -
			*n* = I:91, C:91	* Depressive symptoms (Zung Self-Rating Depression Scale)	15 m: -
				* Stress-related symptoms (Stress and Crisis Inventory)	15 m: -
				* Quality of life (36-item Short-Form Health Survey)	15 m: +/-
				* Pain related disability (Oswestry Disability Index)	15 m: -
				*Costs*	
				* Work ability (Swedish government insurance company)	15 m: +

				*Experienced quality of care by patients*	
				* Patient satisfaction with assessment (study-specific questionnaire)	15 m: +
24	Taylor-Gjevre	-	RCT	*Health*	
	et al., (2017)			* **Disease activity §** (Disease Activity Score-28)	9 m: -
	[53]		*n* = I:31, C:23	* Quality of Life (EuroQoL-5D)	9 m: -
				* Patient’s global function score (100-mm Visual Analogue Scale, global function)	9 m: -
				*Experienced quality of care by patients*	
				* Satisfaction (9-item visit-specific satisfaction questionnaire)	9 m: -
*Non-randomized trial designs*					
25	Burnham et	2006–2007	Prospective	*Health*	
	al., (2010) [48]		cohort	* Pain intensity (0–10 Numeric Rating Scale)	I-4: 0 m-discharge: +
				* **Pain interference ¥** (Pain Interference Questionnaire)	I-4: 0 m-discharge: +
			*n* = 29		
26	Claassen et al.,	2015–2016	Observational	*Health*	
	(2018) [49]		pilot study	* BMI (weight/length2)	0 m-3 m: -
				* Pain and limitations in functional activities (Western Ontario and McMaster	0 m-3 m: -
			*n* = 107	Universities Osteoarthritis Index, pain and physical functioning subscales)	
				* Illness perceptions (Brief illness Perception Questionnaire)	0 m-3 m: +
				* Physical activity (Short Questionnaire to Assess Physical Activity)	0 m-3 m: -
				*Costs*	
				* **Healthcare Utilization ¥** (self-constructed questionnaire and patient diary pain	0 m-3 m: +
				medication, total number of contacts)	
				*Experienced quality of care by patients*	
				* Patient satisfaction (1-item, with satisfaction with course)	+
27	Plagge et al.,	-	Retrospective	*Health*	
	(2013) [51]		study	* Pain severity and interference (Chronic Pain Grade)	0 m-post: All: +
				* Pain catastrophizing (Pain Catastrophizing Scale)	0 m-post: +
			*n* = 30	* Fear avoidance (Tampa Scale for Kinesiophobia)	0 m-post: +
				* Depressive symptoms (Patient Health Questionnaire)	0 m-post: +
				* **Quality of life ¶** (Center for Disease Control Health-Related Quality of Life	0 m-post: All: +
				Measure)	
				* Satisfaction with life (Satisfaction with Life Scale)	0 m-post: +
**In primary care and between primary care and secondary or tertiary care**					
*Randomized trial designs*					
28	Stoffer-Marx	2012–2014	RCT	*Health*	
	et al., (2018)			* Pain	2 m: -
	[54]		*n* = I:59, C:69	* Health status	2 m: -
				* **Grip strength §**	2 m: +

				*Experienced quality of care by patients*	
				* Satisfaction of patients with their health care	2 m: +
**Between primary care and social care**					
*Randomized trial designs*					
29	Bültmann et	2004–2005	RCT	*Health*	
	al., (2009) [55]			* Pain intensity (2-items from Örebro Musculoskeletal Pain Questionnaire)	3 m: + 6 m: -
			*n* = I:66 C:47	* Functional disability (Oswestry Disability Questionnaire)	3 m:-6 m: -
				*Costs*	
				* **Administrative data on cumulative sickness absence hours §** (The Danish	0–3 m:-3–6 m: -
				National Health Insurance Service Registry)	6 m-1 y:+ 0–6 m: + 0–1 y: +
				* Work status; return to work, full-time/part-time sick leave) (from the Danish	*n.d.*
				National Health Insurance Service Registry)	
				* Cost-benefit analysis; cumulative sickness absence hours, consultations and costs	3 m: +/- 1 y: +/-
				of primary health care utilization, outpatient treatment, hospitalization, and	
				prescribed medications (the Danish National Health Insurance Service Registry, the	
				Danish National Patient Registry, and the Danish National Prescription Registry)	
*Non-randomized trial designs*					
30	Heijbel et al.,	2000–2003	Longitudinal	*Costs*	
	(2013) [56]		design	* **Return to work ¥** (number of days to full- or part-time return to work)	2 y: +
			*n* = 779	*Satisfaction with work by HCPs*	
				* Experiences of driving and implementing a workplace-based rehabilitation	+/-
				intervention	
**Between primary care and secondary or tertiary care and social care**					
*Randomized trial designs*					
31	1: Lambeek et	2005–2009	1: Protocol RCT	*1: Health*	*n.a.*
	al., (2007) [58]			* Pain intensity (10-point Visual Analogue Scale)	
			Intended *n* =	* Functional status (Roland Morris Disability Questionnaire)	
			I:65, C:65	* Quality of life (EuroQoL-5D)	
				*Costs*	
				* **Return To Work §** (sick leave in calendar days during study until full return	
				to work in own or other work, for at least 4 weeks without recurrence)	
				* Total duration of sick leave	
				* Direct (non)-medical costs (diaries)	
				*Experienced quality of care by patients*	
				* Patient satisfaction (Patient Satisfaction with Occupational Health Services	
				Questionnaire)	
	2: Lambeek et	2005–2009	2: RCT	*2: Health*	
	al., (2010) [59]			* Pain intensity (10-point Visual Analogue Scale)	3 m:-6 m:-1 y: -
			*n* = I:66, C:68	* Functional status (Roland Morris Disability Questionnaire)	3 m:-6 m:-1 y: +
				*Costs*	
				* **Return To Work §** (sick leave in calendar days during study until full return to	1 y: +
				work in own or other work, for at least 4 weeks without recurrence)	
				* Total duration of sick leave	1 y: +
	3: Lambeek et	2005–2009	3: RCT	*3: Costs*	
	al., (2010) [60]		*n* = I:66, C:68	* **Duration until sustainable Return To Work §** (sick leave in calendar days during	1 y: +
				study until full return to work in own or other work, for at least 4 weeks without	
				recurrence)	
				* Direct (non)-medical costs (diaries)	
				- Total costs and indirect costs	1 y: +
				- Total direct costs	1 y: -
				- Cost-effectiveness	1 y: +/-
				- Cost-benefit	1 y: +
32	1: Steenstra et	2000–2002	1: Protocol RCT	*1: Health*	*n.a*
	al., (2003) [61]			* Functional status (Roland Morris Disability Questionnaire)	
			Intended *n* =	* Pain intensity (10-point Visual Analogue Scale)	
			I:100, C:100	* Kinesiophobia (Tampa Scale for Kinesiophobia)	
				* Fear of movement (Fear Avoidance Beliefs Questionnaire)	
				* Coping (Pain Coping Inventory Scale)	
				*Costs*	
				* **Return to work in the year after the first day of sick leave §**	
				* Workers use of pain medication and use of medical and alternative medical	
				resources and general health status (EuroQoL-5D)	
				*Experienced quality of care by patients*	
				* Patient satisfaction (short version Patient Satisfaction with Occupational Health	
				Services Questionnaire)	
	2: Anema et	2000–2002	2: Pragmatic	*2: Health*	
	al., (2007) [57]		RCT	* Functional status (Roland Morris Disability Questionnaire)	1 y: -
				* Pain (10-point Visual Analogue Scale)	1 y: -
			*n* = I:27, C:85		
				*Costs*	
				* **Sick leave duration due to LBP §**	1 y: -
**Between primary care and community-based care**					
*Randomized trial designs a qualitative designs*					
33	1: McBeth et	-	1: RCT	1: *Health*	
	al., (2012) [66]		*n* = I:102 C:98	* **Change in health §** (7-point, Clinical Global Impression Change Score)	6 m: + 9 m: +
				* Quality of life (36-item Short-Form Health Survey)	
				- Physical component score	6 m: + 9 m: +
				- Mental component score	6 m:-9 m: -
				* Pain severity (Chronic Pain Grade)	6 m:-9 m: -
				* Mental health (General Health Questionnaire)	6 m:-9 m: -
				* Fear of movement (Tampa Scale for Kinesiophobia)	6 m:-9 m: +
				*Costs*	
				* Cost-effectiveness analysis	6 m:-9 m: -
	2: Bee et al.,	-	2: Qualitative	2: *Health*	
	(2016) [62]		study	* Participants’ illness experiences (patients’ physical and emotional reactions to	+/-
				pain, their rationalization of chronic or unexplained symptoms)	
			*n* = 44		
				*Experienced quality of care by patients*	
				* **Participants’ treatment experiences ¥** (their treatment preferences and the	+/-
				perceived fit between the trial interventions and patient need)	
34	1: Bennell et	2012–2015	1: Protocol	1: *Health*	*n.a.*
	al., (2012) [64]		Pragmatic RCT	* **Average pain in the past week §** (11-point Numeric Rating Scale)	
				* Physical function in past 48 h (Western Ontario and McMaster Universities	
			Intended *n* =	Osteoarthritis Index, physical function subscale)	
			I:67, C:67	* Global rating of change	
				* Change in pain (7-point ordinal scale)	
				* Change in physical functioning (7-point ordinal scale)	
				* Physical activity (Physical Activity Scale for the Elderly, Active Australia Survey,	
				stepping duration and steps per day over 7 consecutive days)	
				* Health-related quality of life (Assessment of Quality of Life Instrument version 2)	
				* Mood (Arthritis Impact Measurement Scale Version 2)	
				* Emotional state (Depression, Anxiety and Stress Scale)	
				* Fear of injury (Brief Fear of Movement Scale)	
				* Symptom severity (Patient Health Questionnaire)	
				* Coping (Coping Strategies Questionnaire)	
				* Catastrophizing (Pain Catastrophizing Scale)	
	2: Hinman et		2: Process	2: *Experienced quality of care by patients*	
	al., (2016) [65]		evaluation	*Satisfaction with care by HCPs*	
				- **Theme 1: genuine interest and collaboration**	+
			*n* = 6 pt	- **Theme 2: information and accountability**	+/-
			*n* = 14 HCP	- **Theme 3: program structure**	+/-
				- **Theme 4: roles and communication in teamwork**	+/-
	3: Bennell et	2012–2015	3: Pragmatic	3: *Health*	
	al., (2017) [63]		RCT	* **Knee pain intensity §** (11-point Numeric Rating Scale)	6 m:-1 y:-18 m: -
				* **Physical function in the previous 48 h §** (Western Ontario and McMaster	6 m:-1 y:-18 m: -
			*n* = I:84 C:84	Universities Osteoarthritis Index)	
				* Pain on walking in the past week (11-point Numeric Rating Scale)	6 m:-1 y:-18 m: -
				* Pain (Western Ontario and McMaster Universities Osteoarthritis Index)	6 m:-1 y:-18 m: -
				* Health related quality of life (Assessment of Quality of Life Instrument version 2)	6 m:-1 y:-18 m: -
				* Physical activity (stepping duration and steps per day over 7 consecutive days)	6 m: + 1 y: + 18 m: +
				*Costs*	
				* Number of physiotherapy visits	*No comparison between groups*

**-**: year not known; pt: patient; HCP; healthcare professional; N: number of participants; I: intervention; C: control; §: primary outcome as described in article; ¢: primary outcome as based on sample size calculation; ¥; primary outcome as based on aim of intervention; ¶: primary outcome based on appearance in Quadruple Aim; in bold: primary outcome; n.a.: not applicable; n.d.: no data presented; In randomized designs: + is significant compared to control group; - is non-significant compared to control group; In non-randomized designs: + is significant over time;-is non-significant over time; qualitative: + only positive opinions mentioned; - only negative opinions mentioned; +/- positive as well as negative opinions mentioned. Articles without results are described in grey.

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
