# Peer review of "Interdisciplinary Care Networks in Rehabilitation Care for Patients with Chronic Musculoskeletal Pain: A Systematic Review"

_jcm, 2021, doi:10.3390/jcm10092041_

Round 1
Reviewer 1 Report
You have reviewed pain management interventions which is very relevant on many socio-ecological levels. The review is very comprehensive and done according to the PRISMA systematic review guidelines. Your excellent review is obscured because it is difficult to read. I suggest you work with someone at the writing center at your university or a good English language editor. Although I haven't commented extensively on the writing style, I have attached my PDF file.

Reviewer 2 Report
- This is a systematic review of interdisciplinary care networks related to chronic musculoskeletal pain using PRISMA criteria with appropriate search strategy and appraisal criteria.
- The tables are complicated and landscape formatting is recommended to improve readability, especially tables 2, 3, and the appendix.
- Tables 4-6 are missing, although described in the results. Please add these to the revision.
- The authors appear to mix scientific reporting and APA styles for in-text citations. Numbered scientific reporting is preferred.
- In the discussion the risk of bias is highlighted for references 48, 51, 54 but the details could not be found in the results section. Briefly give the reasons for bias in the discussion. Please include a more detailed discussion of bias for the studies in the results section.
- The final paragraph of the discussion describes a proposal to address a key gap in the knowledge base "Therefore, we propose to develop a core outcome set an assessment method for each Quadruple Aim Goal." Please give more details/suggestions.
Reviewer 3 Report
The manuscript addresses a systematic review of the interdisciplinary care networks in rehabilitation care for chronic musculoskeletal pain patients. Specifically, two research questions are considered. The first, which interdisciplinary networks of primary care for adult chronic musculoskeletal pain patients have been developed in the last 25 years? The second, what has been its impact in terms of "Quadruple Aim outcomes" (health; health care costs; quality of care experienced by patients; work satisfaction for health care professionals)?
The authors have done an exhaustive systematic review, fulfiling all the quality standards according to the PRISMA guidelines. In my opinion, the manuscript is relevant and worthy to be published in JCM. Here are some comments for the authors to consider.
- My main concern is in Table 3. Its information is essential, as it answers the second research question. However, it is difficult to read because of two reasons: corresponding to the “Quadruple Aim outcomes”, up to four outcomes are included in each row; and they also include the full names of the tests within the cells. That is why the table occupies up to 13 pages. I would propose to divide the table into four, one for each of the aims; and abbreviate the names of the tests, placing the full name in a legend at the bottom of the page.
- Likewise, I cannot find tables 4, 5 and 6, referring to the risk of bias, and which are mentioned in the text.
- Finally, in terms of keywords, I think that "chronic musculoskeletal pain" should be included.
Minor Comments
- Why did not the authors use the PRISMA 2020 guide instead of the 2009 ones?
- Is Table 1 really needed? If the authors justify their maintenance, they should introduce the word "inclusion" instead of "in-" in the title.
- Why are the literature searches included in the manuscript when they are already in a supplemental document?
Round 2
Reviewer 1 Report
Thank you, you have made the corrections requested.
Author Response
Thanks for your time to review the manuscript and your valuable comments
Reviewer 2 Report
- For Figure 5 please give a brief explanation for different levels of bias in a legend.
- Lines 686-689 - add more detail of the core outcome management set - maybe: example tools, data collection techniques, definitions
Author Response
We would like to thank reviewer 2 for his/her comments
Comment 1: For Figure 5 please give a brief explanation for different levels of bias in a legend.
Thanks for the comment. In figure 5 the results of the critical appraisal assessing the trustworthiness, relevance and results of published qualitative studies are presented. In case at least 7 of the 10 questions was answered as yes was scored as positive. In other cases the study was scored as negative. We have added the background of the critical appraisal to the methods section (line 211-212). With this appraisal, no different levels of bias are assessed. We gave now a brief explanation (positive or negative methodological quality).
Comment 2: Lines 686-689 - add more detail of the core outcome management set - maybe: example tools, data collection techniques, definitions
We have added a bit more details regarding standardizing measurement moments. However, it is beyond the scope of this review to propose tools or techniques for this outcome set. Future research is needed to define this outcome set.